J Physiol 600.20 (2022) pp 4421–4438

## TECHNIQUES

# THRIFTY: a novel high-throughput method for rapid fibre type identification of isolated skeletal muscle fibres

Oscar Horwath[1] ![ID], Sebastian Edman[1], Alva Andersson[1] ![ID], Filip J. Larsen[1] and William Apró[1,2] ![ID]

[1]*Department of Physiology, Nutrition and Biomechanics, Åstrand Laboratory, Swedish School of Sport and Health Sciences, Stockholm, Sweden*
[2]*Department of Clinical Science, Intervention and Technology, Karolinska Institutet, Stockholm, Sweden*

Handling Editors: Scott Powers & Michael Hogan

The peer review history is available in the Supporting Information section of this article (https://doi.org/10.1113/JP282959#support-information-section).

**Abstract**   Fibre type-specific analyses are required for broader understanding of muscle physiology, but such analyses are difficult to conduct due to the extreme time requirements of dissecting and fibre typing individual fibres. Investigations are often confined to a small number of fibres

**Oscar Horwath** and **Sebastian Edman** are currently PhD-students in the Åstrand Laboratory at the Department of Physiology, Nutrition and Biomechanics, Swedish School of Sport and Health Sciences. They both received their Master's degrees in exercise biomedicine from Halmstad University. Oscar further received a Master's degree from Örebro University in physiology and medicine. His research focuses on the underlying mechanisms of muscle loss in the context of ageing and obesity. Sebastian's research focuses on anabolic and autophagy signalling as well as mitochondrial function in the different human skeletal muscle fibre types.

O. Horwath and S. Edman contributed equally to this work.

from few participants with low representativeness of the entire fibre population and the participant population. To increase the feasibility of conducting large-scale fibre type-specific studies, a valid and rapid method for high-throughput fibre typing of individually dissected fibres was developed and named THRIFTY (for high-THRoughput Immunofluorescence Fibre TYping). Employing THRIFTY, 400 fibre segments were fixed onto microscope slides with a pre-printed coordinated grid system, probed with antibodies against myosin heavy chain (MyHC)-I and MyHC-II and classified using a fluorescence microscope. The validity and speed of THRIFTY was compared to a previously validated protocol (dot blot) on a fibre-to-fibre basis. Fibre pool purity was evaluated using 'gold standard' SDS-PAGE and silver staining. A modified THRIFTY-protocol using fluorescence western blot equipment was also validated. THRIFTY displayed excellent agreement with the dot blot protocol, $\kappa = 0.955$ (95% CI: 0.928, 0.982), $P < 0.001$. Both the original and modified THRIFTY protocols generated type I and type II fibre pools of absolute purity. Using THRIFTY, 400 fibres were typed just under 11 h, which was approximately 3 times faster than dot blot. THRIFTY is a novel and valid method with high versatility for very rapid fibre typing of individual fibres. THRIFTY can therefore facilitate the generation of large fibre pools for more extensive mechanistic studies into skeletal muscle physiology.

(Received 25 April 2022; accepted after revision 25 August 2022; first published online 7 September 2022)

**Corresponding author** W. Apró: Department of Physiology, Nutrition and Biomechanics, Åstrand Laboratory, Swedish School of Sport and Health Sciences, Stockholm, Sweden, Box 5626. SE 114 86 Stockholm, Sweden. Email: william.apro@gih.se

**Abstract figure legend** Large scale investigations into fibre type-specific physiology are limited by the extreme time requirements related to classifying individually isolated muscle fibres. To increase the feasibility of such studies, we developed a rapid and reliable fibre-typing method which we named THRIFTY. Additional benefits of THRIFTY, beyond high speed and validity, include high versatility, excellent scalability and low cost. The many benefits of THRIFTY increase the feasibility of generating large pools of pure type I and type II muscle fibres which can be used for multiple downstream analyses. The high speed of THRIFTY also enables time-sensitive assays where measurements need to be carried out in close connection with tissue sampling. Employing THRIFTY may therefore provide new insights into fibre type-specific muscle physiology which may have broad implications in health and disease.

## Key points

- Skeletal muscle is composed of different fibre types, each with distinct physiological properties.
- To fully understand how skeletal muscle adapts to external cues such as exercise, nutrition and ageing, fibre type-specific investigations are required.
- Such investigations are very difficult to conduct due to the extreme time requirements related to classifying individually isolated muscle fibres.
- To bypass this issue, we have developed a rapid and reliable method named THRIFTY which is cheap as well as versatile and which can easily be implemented in most laboratories.
- THRIFTY increases the feasibility of conducting larger fibre type-specific studies and enables time-sensitive assays where measurements need to be carried out in close connection with tissue sampling.
- By using THRIFTY, new insights into fibre type-specific muscle physiology can be gained which may have broad implications in health and disease.

## Introduction

Skeletal muscle exhibits plasticity and is a heterogeneous tissue composed of different fibre types, each with distinct contractile and metabolic properties (Schiaffino & Reggiani, 2011). Historically, different muscles were broadly distinguished based on their colour due to their varying myoglobin content (Needham, 1926). However, over the years, more sophisticated biochemical methods were developed, and today muscle fibres are categorized mainly according to their myosin heavy chain (MyHC)-isoform expression (Schiaffino et al., 1985). Adult human skeletal muscle is composed primarily of three distinct fibre types (type I, type IIA and type IIX)

(Schiaffino, 2010; Schiaffino & Reggiani, 2011), with a minor presence of a fibre pool that co-express two or more MyHC isoforms, commonly referred to as hybrid or intermediate fibres (Medler, 2019).

Different fibre types exhibit considerable diversity in terms of contractile characteristics and physiological properties such as glycogen utilization and metabolite production (Esbjornsson-Liljedahl et al., 1999; Soderlund et al., 1992). The existing heterogeneity amongst fibre types is reflected also in their ability to adapt to external stimuli (Edman et al., 2019; Horwath et al., 2020; Tannerstedt et al., 2009; Tesch, 1988; Verdijk et al., 2007). Given the large inter-individual variation in fibre type composition of human muscle (Horwath et al., 2021; Lexell et al., 1983), and its plasticity in response to different muscle loading patterns (Andersen & Aagaard, 2000; Staron et al., 1990), conducting investigations in a fibre type-specific manner to control for this variability is warranted.

To date, numerous studies have examined physiological variables such as glycogen utilization, metabolite production, mRNA expression and protein abundance in different fibre types from human skeletal muscle. In these studies, analyses have been performed on individually isolated muscle fibres, or on pools of fibres of the same type. However, due to the cumbersome nature of isolating individual muscle fibres, many studies have included relatively few fibres, some as few as two fibres per tissue sample studied (Wang & Sahlin, 2012).

It is well established that some physiological properties are fibre type-specific, but it must also be noted that variation exists not only between fibre types, but also between individual fibres within the same type. Considerable variability has been reported for glycogen utilization (Essen & Henriksson, 1974; Vollestad et al., 1984), metabolite content (ATP, NADH and PCr) (Ren et al., 1988; Sahlin et al., 1997; Soderlund & Hultman, 1990), and protein phosphorylation (Edman et al., 2019), within the same fibre type. While the exact number of fibres required is unclear, these variations stress the importance of including a large number of fibres to capture a representative portion of the entire fibre type population. Beyond the issue of poor representativeness, some techniques, such as muscle protein synthesis measurements using stable isotope tracers, may require a relatively large sample mass for proper resolution and accuracy. This, in turn, may be difficult to obtain with a low number of fibres.

Fibre type-specific aspects of muscle physiology are most commonly addressed by examining labelled muscle cross sections (Tobias & Galpin, 2020) or preparations of either individual muscle fibres (Edman et al., 2019; Wyckelsma et al., 2015) or pools of muscle fibres (Skelly et al., 2021; Tannerstedt et al., 2009). While the latter two

are most versatile in terms of analytical techniques that can be utilized, they are also the most time demanding. A major challenge in large scale fibre type-specific analyses is related to the methods currently available for fibre type classification of individually isolated fibres. The gold standard, and the method most commonly used, is single fibre (sf)SDS-PAGE (Tobias & Galpin, 2020), where fibre type identification is based on the differential migration of the various MyHC isoforms through the gel due to differences in molecular mass. Fibre typing can also be performed by employing classical western blot analysis of single fibre preparations (sfWB) where fibre type identification is achieved with MyHC-specific antibodies (Edman et al., 2019; Murphy, 2011). However, both sfSDS-PAGE and western blotting are time-consuming and resource-intensive methods with less than optimal practicality for large scale fibre typing.

A significant hurdle in large scale fibre type-specific research on isolated muscle fibres is the dissection process itself. Even when performed by the most skilled technician, dissecting thousands of individual fibres will always take a considerable amount of time. However, beyond skill, there is little room for technical improvement of the dissection process. As such, the only viable option for increasing the feasibility of larger scale studies on isolated muscle fibres is to improve fibre typing efficiency. In this regard, Christiansen et al. (2019) recently presented a method in which they adapted the traditional dot blot technique for single fibre typing using MyHC-specific antibodies (Christiansen et al., 2019). When compared to single fibre western blot, the dot blot technique was shown to be a valid method with increased time efficiency. However, the original protocol limits the applicability of the method to a single analysis, i.e. western blot, and if the method was to be adapted for multiple analyses, much of the gain in time efficiency would be lost. Thus, despite the significant increases in time efficiency presented by Christiansen et al. (2019), there remains a need for further methodological developments to facilitate large scale fibre typing, especially if multiple analytical techniques are to be used on the same sample.

Building on the method originally described by Essen et al. (1975), we have developed a versatile, high-throughput fibre typing method based on immunofluorescence detection with high validity. The method, which we have named THRIFTY (for high-THRoughput Immunofluorescence Fibre TYping), allows for fibre typing at superior speeds compared to other contemporary methods and is considerably less resource demanding. Application of THRIFTY may provide unique opportunities to gain deeper mechanistic insights in fibre type-specific muscle physiology and fibre type-specific adaptations to external stimuli.

## Methods and results

### Ethical approval

Muscle samples used in this study were obtained from three ongoing research projects which have received ethical approval from the Swedish Ethical Review Authority (DNR 2017/2107-31/2, DNR 2019-0038 1 and DNR 2017/2034-31/2) and are in agreement with the *Declaration of Helsinki*, apart from registration in a database. Participants gave oral and written consent after being informed about the procedures and possible risks of each study.

### Tissue sampling and isolation of individual muscle fibres

Muscle biopsies were collected from four healthy, physically active, non-smoking participants: two young females (29 and 30 years), one young (30 years) and one older male (65 years). Muscle biopsies were collected from the vastus lateralis following administration of local anaesthesia (Carbocain 20 mg ml$^{-1}$, AstraZeneca AB, Södertälje, Sweden) using a Bergström needle with manually applied suction (Ekblom, 2017). Three of the samples were immediately blotted free of blood, frozen in liquid nitrogen and stored at $-80°C$. These tissue samples were then lyophilized overnight and subsequently dissected free of blood and connective tissue, leaving intact fibre bundles. From these, individual muscle fibres were isolated using needles and fine forceps (Fig. 1). All dissections on lyophilized muscle were carried out at room temperature in a climate-controlled room ($\leq 40\%$ humidity). One sample was used for dissection under wet conditions. Immediately after collection, the biopsy sample was placed in a Petri dish filled with ice-cooled biopsy preservation medium (BIOPS; 10 mM Ca-EGTA, 0.1 $\mu$M free Ca, 20 mM imidazole, 20 mM taurine, 50 mM K-MES, 0.5 mM dithiothreitol, 6.56 MgCl$_2$, 5.77 mM ATP, 15 mM phosphocreatine, pH 7.1) on top of a bed of ice. Dissection of both lyophilized and wet fibres was carried out under a stereo microscope (VisiScope, SZB260, VWR International AB, Stockholm, Sweden).

### Overview of THRIFTY

Employing the THRIFTY method, a minimal segment ($\sim$0.5 mm) was cut from individual fibres and mounted on a microscope slide (cat. no. 631-1554, VWR International). The microscope slide was customized with a 10 × 20 square grid system printed with white, solvent-resistant ink at a line thickness of 0.5 mm. Each square measured 1.05 × 1.05 mm and could be identified through a coordinate annotation system (Fig. 2). Printing of the square grid was performed by a commercial

printing company (Creative Reklam i Sverige AB, Ludvika, Sweden).

Immediately after placing a segment in a square, it was fixed to the slide by applying a small drop of distilled water onto the segment with a Hamilton syringe and letting it dry fully. For the lyophilized fibres, a square grid on a sheet of black A4 paper, identical to that on the slide, was used to keep track of each individual fibre for the purpose of pooling after typing was completed. Thus, after the small segment had been fixed to the slide, the remaining piece of the fibre was placed in the corresponding square on the paper, e.g. for a segment placed in square A0 on the slide, the remaining fibre was placed on square A0 on the paper, etc. For the wet fibres, the remaining piece of the fibre was placed in biopsy preservation medium in individual wells of a 96-well plate. Once all segments were dry, the microscope slides were submerged in a polypropylene slide mailer (similar to a Coplin jar) with acetone for 3 min to permeabilize the fibre segments. After permeabilization, the slides were removed from the acetone and left to dry ($\sim$ 2–3 min). Once the acetone had evaporated, the slides were submerged in a separate slide mailer containing antibody solution with primary antibodies directed against MyHC-I (BA-F8-s, 1:100) and MyHC-II (SC-71-s, 1:50), diluted in phosphate-buffered saline (PBS) containing 5% normal goat serum (NGS, Thermo Fisher Scientific, Waltham, MA, USA) and 1% Triton X-100. The primary antibodies were purchased

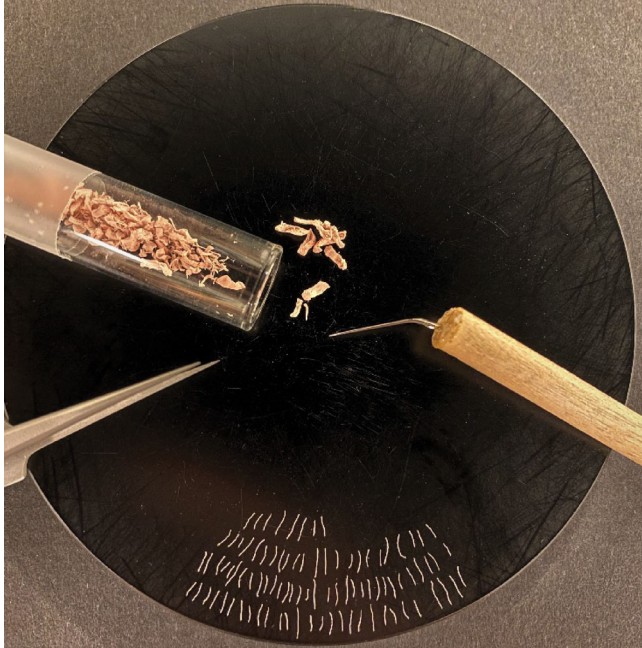

**Figure 1. Image of needles and fine tweezer used to dissect bundles of freeze-dried muscle**
Isolated individual muscle fibres are lined up in the bottom of the cutting board. [Colour figure can be viewed at wileyonlinelibrary.com]

from Developmental Studies of Hybridoma Bank (DSHB) (Iowa City, IA, USA) and have been used extensively to identify type I and type II fibres of human skeletal muscle (Bloemberg & Quadrilatero, 2012; Murach et al., 2019; Smerdu & Soukup, 2008). Following primary antibody incubation, the slides were washed by sequentially submerging them into three separate slide mailers containing PBS, each time for 5 min. Next, the slides were submerged in a separate slide mailer containing secondary antibody solution with species- and subclass-specific secondary antibodies diluted in PBS containing 1% NGS and 1% Triton X-100 and incubated for 30 min in the dark. The secondary antibodies were Alexa-Fluor goat anti-mouse IgG2b 488 (1:1000) and goat anti-mouse IgG1 647 (1:1000) (purchased from Thermo Fisher Scientific). Finally, slides were washed as described above and subsequently mounted with ProLong Gold anti-fade reagent (Thermo Fisher Scientific) and a coverglass. Importantly, to minimize the risk of fibre segments detaching from the slides, all the above steps involving transfer of slides to new solutions were performed by carefully submerging the slides into mailers pre-filled with each solution, and never by pouring the solutions into the mailers containing the slides.

Fibre type identification was performed using a digital fluorescence microscope (CELENA S Digital Imaging System, Logos Biosystems, Anyang, South Korea) equipped with LED filter cubes for fluorophore visualization; an enhanced yellow fluorescent protein (EYFP) filter (Ex500/20, Em535/30) for Alexa-Fluor 488 (conjugated to the secondary antibody against BA-F8-s, MyHC-I) and a Cy5 long pass filter (Ex620/60, Em665lp) for Alexa-Fluor 647 (conjugated to the secondary antibody against SC-71-s, MyHC-II). Each square of the grid was sequentially viewed in each channel at ×4 magnification and image settings were identical for light (100%) and gain (18 dB) but differed for exposure time (Alexa-Fluor 488, 40 ms; Alexa-Fluor 647, 10 ms). Each fibre was then categorized either as type I or type II based on the staining intensity in the two different channels, i.e.

BA-F8 positive and SC-71 negative fibres were classified as type I while BA-F8 negative and SC-71 positive fibres were classified as type II (see Fig. 3). Each fibre had to display high staining intensity at the ends of the fibre segment and within the intracellular compartments (see example in Fig. 4). In cases in which a segment manifested low uniform staining throughout, it was examined at higher magnifications (×10 and ×20) to exclude the possibility of a hybrid fibre type. Fibre segments that showed clear signs of specific binding in both channels (BA-F8 positive and SC-71 positive) were classified as hybrid fibres (example in Fig. 3).

A schematic overview of THRIFTY is illustrated in Fig. 5.

## Validation of THRIFTY

Once the protocol was optimized, we proceeded to validate THRIFTY against the dot blot method (Christiansen et al., 2019). This method was chosen for the primary comparison as it was recently shown to be a reliable and fast method for fibre typing individual muscle fibres. We therefore considered the dot blot procedure to be the best method for larger scale fibre typing and therefore the most appropriate method for validating THRIFTY.

For the validation of THRIFTY, 400 lyophilized muscle fibres were dissected out and treated as described above with a few additional steps. Immediately after the small fibre segment had been cut off and fixed to a square on the THRIFTY slide, the entire remaining fibre was carefully placed in a marked 500 $\mu$l Eppendorf tube, corresponding to the square on the slide. Thus, for the two slides used (i.e. 200 fibre segments per slide), the corresponding number of marked Eppendorf tubes was 400. Next the tubes were centrifuged for 15 s at 16,000 *g* to ascertain that the fibre was at the bottom of the tube. Ten microlitres of 1× Laemmli sample buffer (Bio-Rad Laboratories, Hercules, CA, USA) containing 5% $\beta$-mercaptoethanol (Bio-Rad) was then added to each tube, after which the samples

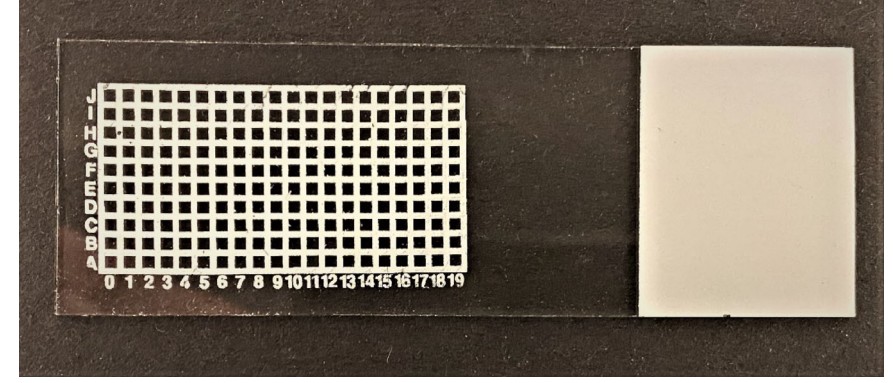

**Figure 2. Image of the microscope slide containing a customized 10 × 20 square grid system printed with white, solvent resistant ink at a line thickness of 0.5 mm**
Each square measured 1.05 × 1.05 mm and could be identified through a coordinate annotation system ranging from A to J vertically and from 0 to 19 horizontally. [Colour figure can be viewed at wileyonlinelibrary.com]

were heated for 5 min at 95°C to denature proteins. After heating, samples were vortexed for 10 s and centrifuged once more for 15 s.

Next, 2 $\mu$l of each sample was dotted onto 10 × 15 cm polyvinylidene difluoride (PVDF)-membranes (Bio-Rad) which immediately prior to the dotting procedure had been activated in methanol. To enable accurate application of each sample during the dotting process, membranes were placed in a transparent tray positioned on top of a 6 × 9 square grid. Thus, a total of 54 samples were dotted on a single membrane. Due to the evaporation of methanol and subsequent drying of the membrane, it was quickly re-activated once after application of approximately half the samples.

Upon completion of the dotting procedure, membranes were once again re-activated in methanol due to evaporation, and then washed in Tris-buffered saline (TBS) for 3 × 3 min. Next the membranes were blocked for 30 min at room temperature in TBS containing 5% non-fat dry milk after which they were incubated with separate primary antibodies for 60 min at room temperature targeting either MyHC-I or MyHC-II. The same primary antibodies were used as for THRIFTY, but concentrate instead of supernatant, i.e. BA-F8-c for MyHC-I and SC-71-c for MyHC-II. Both antibodies were diluted 1:1000 in TBS containing 2.5% fat-free milk and 0.01% Tween-20 (TBS-T). Following washes in TBS-T, membranes were incubated for 45 min with a horseradish peroxidase-conjugated anti-mouse secondary antibody (cat. no. 7076, Cell Signaling Technology, Danvers, MA, USA) diluted 1:10,000 in TBS-T with 2.5% milk. Finally, membranes were incubated with chemiluminescent substrate (SuperSignal West Femto, Thermo Fisher Scientific) for 5 min before imaging. Importantly, since chemiluminescence detection is a single-channel detection method, i.e. it only detects light

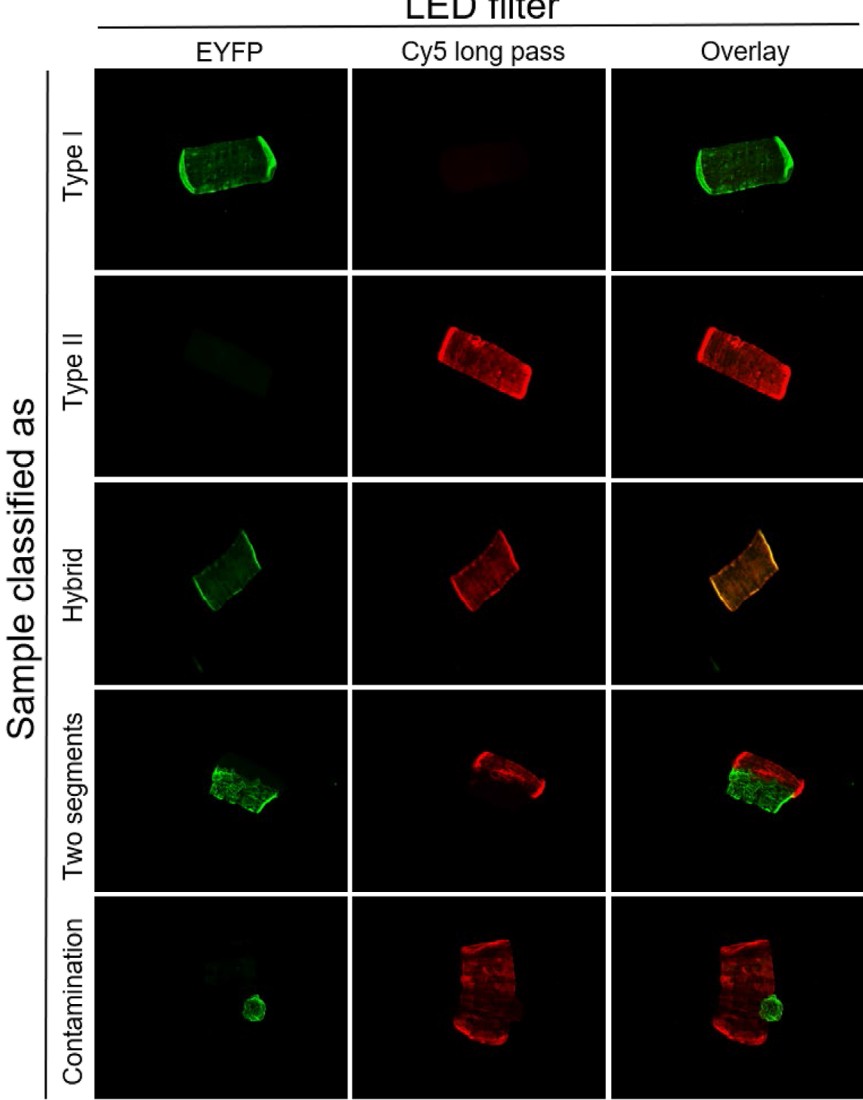

**Figure 3. Image of representative fluorescence staining using THRIFTY on lyophilized fibres**
Samples were classified as pure type I or type II if segments were free of contamination and displayed high fluorescence signal in fibre ends and in the intracellular compartment in each respective filter. Hybrid segments displayed high fluorescence signal in fibre ends and in the intracellular compartment in both filters. If two or more segments were present in the same square they were classified as contaminated. [Colour figure can be viewed at wileyonlinelibrary.com]

at a single wavelength, it cannot be used for detection of separate signals from different antibodies in the dot blot method. Therefore, every sample was dotted on two separate membranes for separate but parallel incubation with either BA-F8-c or SC-71-c as described above.

Image capture was carried out using the ChemiDoc MP imaging system (Bio-Rad) in chemiluminescence mode with 300 s exposure time and $8 \times 8$ binning. Fibre type identification was performed on merged images of the two membranes (see Fig. 6), where dots that were BA-F8 positive and SC-71 negative were classified as type I fibres whereas BA-F8 negative and SC-71 positive were classified as type II fibres. Dots that were positive for both antibodies were considered contaminated.

Out of the 400 segments analysed, seven were excluded due to either detachment during the THRIFTY staining protocol ($n = 4$) or lack of signal with the dot blot method ($n = 3$). With the dot blot method, 171 fibres were identified as BA-F8 positive while 214 fibres were identified as SC-71 positive. Eight fibres were considered inconclusive due to immunoreactivity to both antibodies. With the THRIFTY protocol, 170 fibres were identified as BA-F8 positive and 216 fibres as SC-71 positive. Seven fibre segments were deemed inconclusive (four hybrid fibres and three with dual fibre segments; see Fig. 3 for

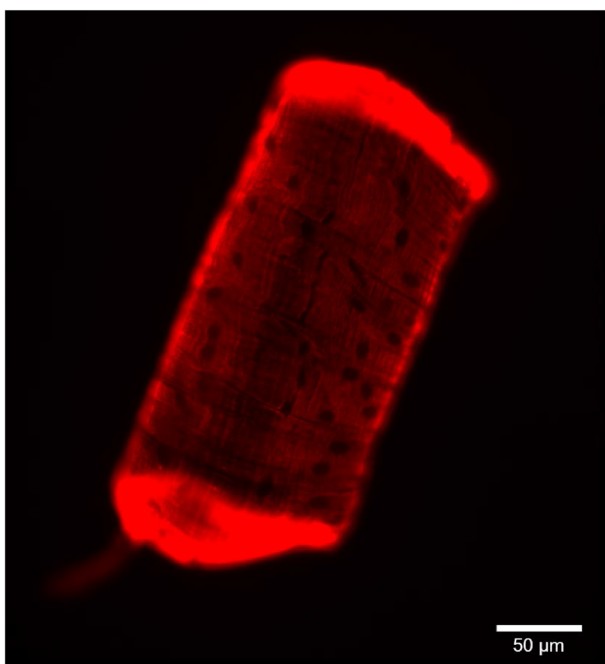

**Figure 4. Image of an individual fibre segment taken at high magnification (×20) using the Cy5 long pass filter**
The segment displays high staining intensity at the ends of the fibre segment and within the intracellular compartments. This fibre segment displayed immunoreactivity for the SC-71 antibody and was thus classified as a type II fibre. The black unstained dots represent myonuclei. [Colour figure can be viewed at wileyonlinelibrary.com]

visual example). A total of nine segments were typed differently between the two methods; five were classified as inconclusive with the dot blot method whilst being either BA-F8 or SC-71 positive in the THRIFTY method, and four segments showed the opposite pattern, i.e. were considered inconclusive with THRIFTY but were either BA-F8 or SC-71 positive in the dot blot method. No fibres were considered pure BA-F8 positive in one method whilst being pure SC-71 positive in the other. Agreement between the two methods was determined with Cohen's $\kappa$ statistical test for inter-rater agreement on the 393 fibres which yielded a detectable signal with both methods. There was a strong agreement between the dot blot and THRIFTY, $\kappa = 0{,}955$ (95% CI: 0.928, 0.982), $P < 0.001$ (IBM SPSS Statistics Version 27, IBM Corp., Armonk, NY, USA).

## Time requirements of THRIFTY *versus* dot blot

As the premise of developing THRIFTY was to facilitate larger scale analysis on pools of type I and type II muscle fibres through increased speed, we compared the time requirements of THRIFTY to that of the dot blot method. For this comparison, 400 fibres were isolated from lyophilized muscle and treated as described above, with some minor modifications.

First, the dissected fibres were individually placed in squares on the gridded black A4 papers described above. Individual fibres were then taken from the square grid and placed on the microscope stage for cutting of the small segment to be fixed on the slide or placed in a marked tube. The remaining piece of the fibre was then returned to its designated square and this procedure was repeated for every individual fibre. To minimize the risk for an order effect, the procedure was carried out in a series of 50 fibres at a time, alternating between the two methods. Ultimately, 400 fibre segments were fixed to two THRIFTY slides, i.e. 200 segments per slide, and 400 segments were placed in 400 individually marked 500 $\mu$l Eppendorf tubes. Next, samples were treated according to each method as described in the sections above.

To reflect the time requirements in a laboratory setting where THRIFTY was used for the first time without previous experience, a novice practitioner performed all the steps described above. Each step was timed for both methods, but for some steps, for practical reasons, time was recorded for a sub-group of the samples and then extrapolated to reflect the whole sample set.

The initial step in THRIFTY, i.e. cutting and fixing 400 segments to two slides, required 6 h 55 min. The staining procedure, which included permeabilization, primary and secondary antibody incubations, washes in-between as well as adding antifade reagent and mounting coverglasses, required 2 h 10 min. The final

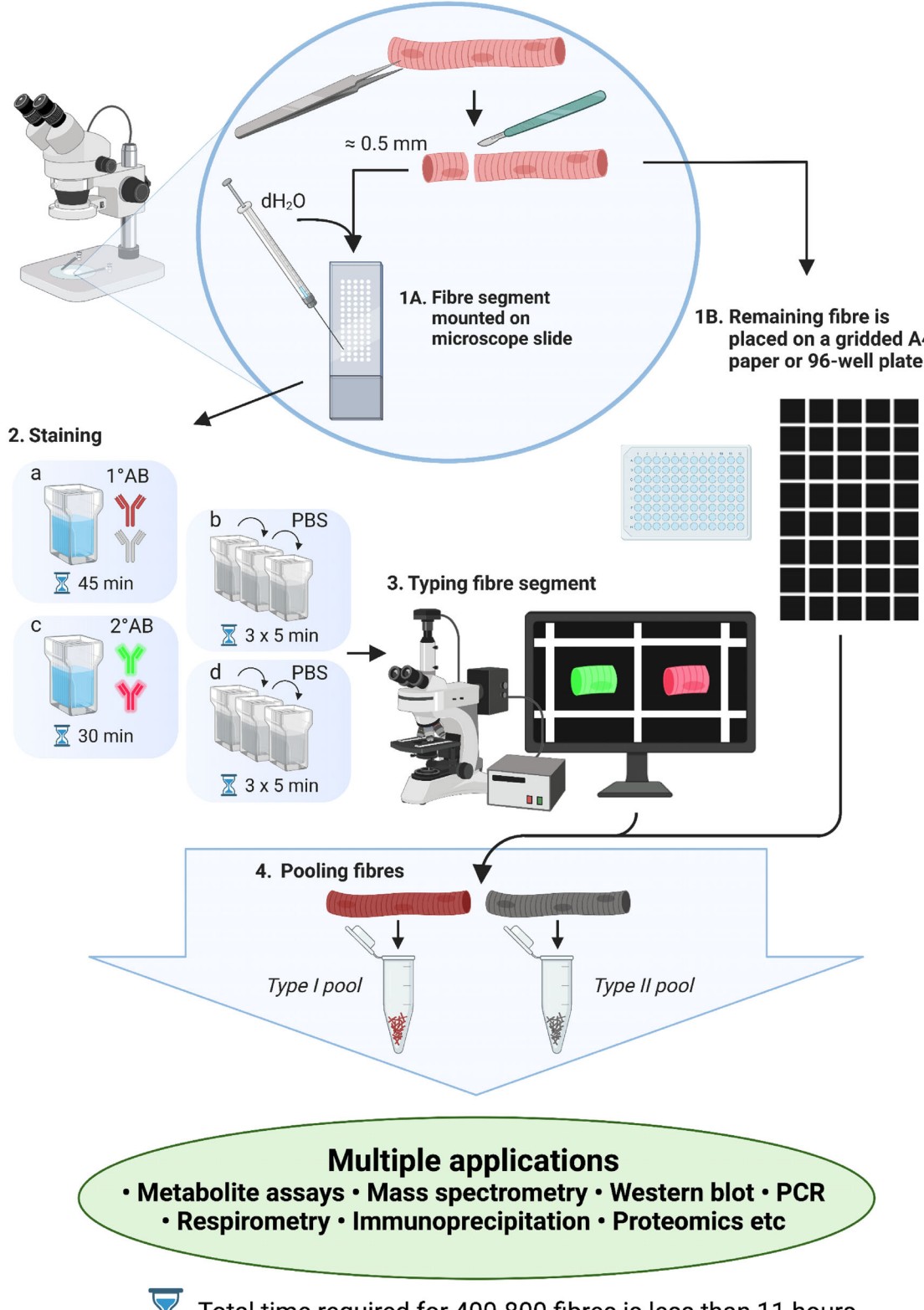

**Figure 5. Schematic overview of the THRIFTY protocol**
Minimal segments (∼0.5 mm) were cut off from individual fibres and mounted on a customized microscope slide containing a printed square grid system. Immediately after placing the segment in a square, it was fixed to the slide by applying a small drop of distilled water on to the segment with a Hamilton syringe and letting it fully dry

(1A). After the small segment had been fixed to the slide, the remaining fibre was placed in the corresponding square on a sheet of black A4 paper, identical to that on the slide (1B), or for wet muscle, in individual wells of a 96-well plate. Once all segments were dry, the microscope slides were submerged in a polypropylene slide mailer (illustrated as Coplin jar in figure) permeabilized in acetone and stained with primary antibodies directed against MyHC-I and MyHC-II (2A). Following washes in PBS (2B), the slides were incubated with species- and subclass-specific secondary antibodies (2C), washed in PBS and mounted with anti-fade reagent and coverglasses (2D). Fibre type identification was then performed using a digital fluorescence microscope (3), after which the fibres (placed on black A4 paper or 96-well plate) were pooled together according to their fibre type (4). [Colour figure can be viewed at wileyonlinelibrary.com]

step of fibre type identification in the fluorescence microscope required 1 h 43 min. Thus, from start to finish, using THRIFTY, 400 muscle fibres were analysed in 10 h 48 min.

For the dot blot method, in addition to 1 h 8 min for marking 400 Eppendorf tubes, cutting and placing 400 segments in these tubes required 8 h 56 min. The sample preparation step, which is not required in THRIFTY, included centrifugation, addition of sample buffer, heating, vortexing and a second centrifugation. Centrifugation as well as heating was performed at maximum capacity for the centrifuge and heating block, i.e. 24 and 48 samples at a time, respectively. The time required for sample preparation was 4 h 20 min, including the time for placing tubes in, and removing tubes from, the centrifuge and heating block.

The next step in the dot blot method was the dotting of samples on PVDF membranes activated in methanol. This was done in duplicate for the separate but parallel incubation with primary antibodies against MyHC-I and MyHC-II (see detailed description in previous section). The time required for dotting of 400 samples (54 samples per membrane) in duplicate (16 membranes in total) was 4 h 9 min. The following staining procedure included re-activation in methanol following dotting of the samples, washing, blocking, primary and

secondary antibody incubations, washes in between, as well as incubation in chemiluminescent substrate and membrane development in the western blot imager. For practical reasons, the staining procedure was performed on four membranes at a time and the total time required for staining all 16 membranes was 17 h 49 min. Lastly, the final step of fibre type identification required 27 min. Thus, from start to finish, using the dot blot method, 400 muscle fibres were analysed in 36 h 49 min.

## Modification of THRIFTY for western blot imaging system

Based on the above, THRIFTY is clearly a valid and rapid method for fibre typing individually dissected muscle fibres. However, its wider applicability may be limited due to its reliance on the use of a fluorescence microscope, equipment which may not be readily available in muscle physiology laboratories. We therefore sought to determine if the final step in THRIFTY, i.e. fibre type identification, could be performed accurately using a western blot imaging system with fluorescence multiplexing capabilities, which is likely more commonly available equipment.

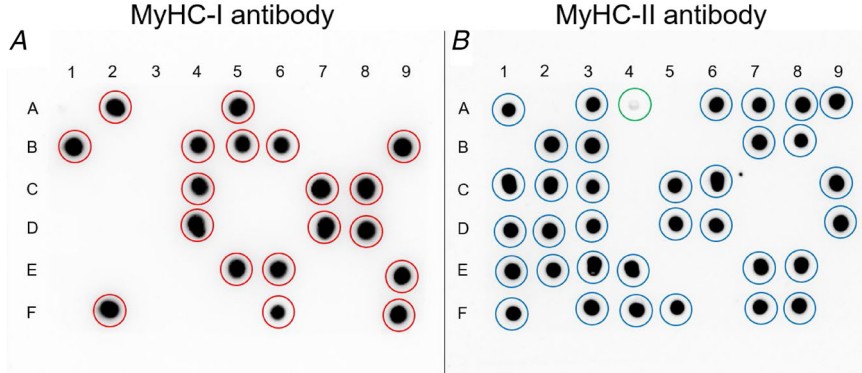

**Figure 6. Fibre segments classified as type I or type II muscle fibres using MyHC-specific antibodies and the dot blot method**
One sample was loaded in the same position on two different membranes (54 samples in total), in which membrane A was probed with the BA-F8 antibody (MyHC-I) and membrane B was probed with the SC-71 antibody (MyHC-II). Red circles indicate segments classified as type I fibres and blue circles indicate segments classified as type II fibres. The green circle located in position A4 on membrane B indicates a segment classified as a type II fibre, but this was only visible after the membrane had been contrasted. [Colour figure can be viewed at wileyonlinelibrary.com]

For this purpose, THRIFTY slides, prepared as described above, were imaged using a ChemiDoc MP imaging system. The slides were imaged using automatic detection in multiplexing mode with Alexa 488 and Alexa 647 channels for identification of BA-F8 (MyHC-I) and SC-71 (MyHC-II), respectively. These initial tests revealed that the signal-to-noise ratio was considerably lower in the ChemiDoc MP compared to the fluorescence microscope, meaning that there was significant fluorescence detected in fibre segments that were known to be BA-F8 and SC-71 negative. This was particularly evident in the Alexa 488 channel which was used for identification of BA-F8 positive segments. In contrast to the fluorescence microscope, it was not possible to confirm or refute the MyHC-isoform based on the staining pattern at the ends of the segment and within the intracellular compartments due to the lower resolution and limited magnification capabilities of the ChemiDoc MP. With fluorescence western blotting, it is known that membrane autofluorescence is more readily detectable at lower wavelengths, but this can be greatly reduced by using secondary antibodies with fluorophores emitting light at higher wavelengths, especially in the near-infrared range. Suspecting autofluorescence, the Alexa-Fluor 488 secondary antibody was therefore substituted for an Alexa-Fluor 790 secondary antibody (goat anti-mouse IgG2b, Jackson ImmunoResearch Europe Ltd, Ely, UK) and multiplex imaging was performed again, but with the Alexa 790 channel for identification of BA-F8 (MyHC-I).

Switching to Alexa-Fluor 790 improved the signal-to-noise ratio to such an extent that it was deemed likely to be suitable for fibre typing. To determine if this was indeed the case, a 'control slide' was prepared with segments from fibres pre-classified as type I or type II using the original THRIFTY protocol. The idea was to see if differences in signal intensity in the two different channels would be distinct enough to enable fibre type classification through visual assessment of the fibre segments. To prepare the control slide, 100 type I and 100 type II fibre segments were fixed to the slide in alternating columns with 10 fibre segments of each type. The slide was then treated according to the THRIFTY protocol, with the modifications that Alexa-Fluor goat anti-mouse IgG2b 790 (1:2000) was used to detect the BA-F8 antibody, and that the application of anti-fade reagent and mounting of the coverglass was omitted. Once the slide had dried completely after the last washing step, the slide was imaged in multiplexing mode with exposure settings in auto-optimal.

Visual assessment of the images identified clear differences in signal intensity between type I and type II fibres (see Fig. 7*A*), which indicated that this method might indeed be suitable for fibre typing. To provide an objective measure of the differences in signal intensity

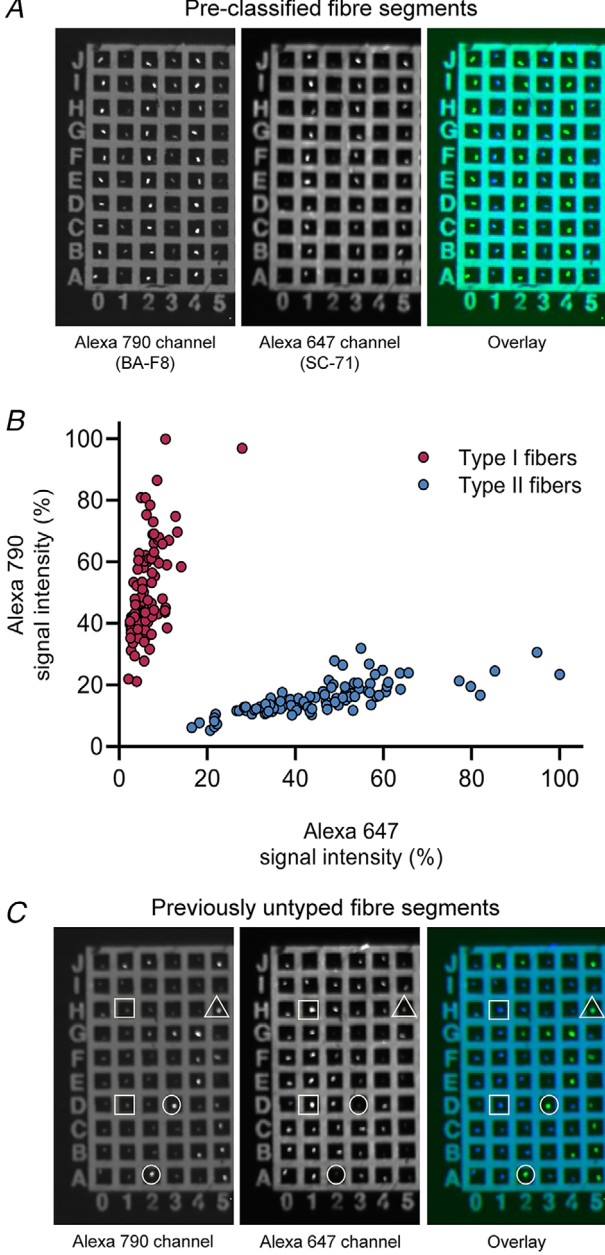

**Figure 7. Modification of THRIFTY for western blot imaging system**
*A*, cropped image of pre-classified fibre segments on the 'control slide' captured using ChemiDoc MP. Type I and type II fibre segments mounted in an alternating pattern are visualized in the Alexa 790 and the Alexa 647 channel, respectively. *B*, the signal intensity of each fibre segment mounted on the 'control slide' expressed as a percentage of the highest value corrected for background fluorescence. *C*, cropped image of previously untyped fibre segments captured using ChemiDoc MP. A random pattern of type I and type II fibre segments was seen in the Alexa 790 and the Alexa 647 channel, respectively. Type I fibre segments are denoted as circles (●), type II fibre segments as squares (■), and inconclusive fibre segment as triangles (▲). [Colour figure can be viewed at wileyonlinelibrary.com]

and exclude the risk of bias in the visual assessment due to the pre-typing procedure, the signal intensity of each fibre segment was quantified. Segments were encircled using the free hand tool in Image Lab software (version 6.1, Bio-Rad) and signal intensity was expressed as a percentage of the highest value corrected for background fluorescence using local background subtraction. As shown in Fig. 7*B*, there was a clear divergence in signal intensity between the two channels. However, while the overall pattern was distinctly different between the two channels, some overlap existed between channels for some of the fibre segments, meaning that the background fluorescence in the 'negative' channel for some fibres was in the same range as the fluorescence signal in the 'positive' channel of other fibres. This illustrates the need to assess the relative difference in signal intensity between channels for each fibre segment.

To determine the applicability of visual assessment in the ChemiDoc MP, segments from 200 previously untyped individual fibres were fixed to a THRIFTY slide and subsequently treated and imaged as described above. All segments were examined in one channel at a time and noted as positively or negatively stained based on the signal intensity in each channel. Segments were then crosschecked between channels (see Fig. 7*C*), and if the same segment was designated as positively stained in both channels the staining was noted as inconclusive. Of the 200 segments, one had detached and out of the remaining 199 segments, 55 were classified as BA-F8 positive (MyHC-I), 138 as SC-71 positive (MyHC-II) and six as inconclusive.

### Validation of fibre pool purity using SDS-PAGE

Finally, to determine the validity of the modified THRIFTY protocol using the ChemiDoc MP imaging system, fibres of the same type were pooled together and subjected to SDS-PAGE and silver staining as described previously (Horwath et al., 2021). All 55 fibres classified as BA-F8 positive were pooled together for the type I fibre pool and an equal number of randomly selected SC-71 positive fibres were included in the type II fibre pool. For comparison, this approach was also carried out on pools of type I and type II fibres with an equal number of randomly selected freeze-dried fibres classified using the original THRIFTY protocol and the dot blot method, as well as 36 type I and 42 type II fibres isolated from wet muscle classified using the original THRIFTY protocol. As shown in Fig. 8, each pool displayed absolute purity regardless of which protocol was used for fibre type identification prior to pooling. Thus, the modified THRIFTY protocol employing a western blot imaging system with fluorescence capabilities (ChemiDoc MP) is a valid and reliable alternative to produce pure pools of type I and type II muscle fibres if a fluorescence microscope cannot be used.

### Applications

To illustrate real-life applicability of THRIFTY in generating large pools of fibres for multiple downstream analyses, 400 muscle fibres were isolated from two muscle biopsies (i.e. 200 fibres from each biopsy) collected from the vastus lateralis muscle of each leg from a 30-year-old male participant in an ongoing study (DNR 2019-0038 1). Both biopsies were collected in the fasted state. Prior to biopsy collection, the participant completed a glycogen loading protocol followed by single legged cycling to deplete glycogen levels in one leg. Thus, on the morning of biopsy sampling, one leg was glycogen loaded while the contralateral leg was glycogen depleted.

Muscle fibres were classified using the original THRIFTY protocol and fibres of the same type from each biopsy were pooled and subsequently weighed on an ultra-micro balance with readability of 0.1 $\mu$g (Cubis MCA2.7S-2S00-M, Sartorius Lab Instruments GmbH & Co, Göttingen, Germany). The number of fibres in each pool ranged from 83 to 109 and the weight of each pool ranged from 423.4 to 644.5 $\mu$g.

Each pool was then homogenized (Apro et al., 2015) and aliquoted for subsequent analysis of (1) citrate synthase activity using a spectrophotometric assay (Alp et al., 1976), (2) muscle glycogen content using a fluorometric assay (Harris et al., 1974; Lowry & Passonneau, 1972), (3) muscle amino acid concentrations using ultra-performance liquid chromatography (Jonsson et al., 2022), and (4) total expression of selected proteins employing western blot (Apro et al., 2015). See specific references for more details. All analyses were performed on whole homogenates containing 10 $\mu$g muscle except total protein expression which was performed on cleared lysates obtained after centrifugation for 10 min at 10,000 $g$. In the western blot analysis, 1.5 $\mu$g protein from the cleared lysate was loaded into each well and total expression of the following proteins was measured in

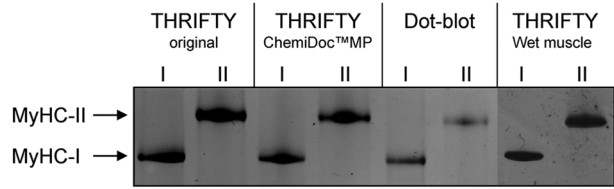

**Figure 8. Image of the gel obtained after MyHC separation by SDS-PAGE showing pooled fibre segments typed as either type I or type II fibres from the original THRIFTY protocol, the modified THRIFTY protocol using ChemiDoc MP, the dot blot method and the THRIFTY protocol using wet muscle**
The gel has been cropped to display protein targets of interest and contrasted for increased visualization of band intensity.

each pool of fibres: mechanistic target of rapamycin (cat. no. 4517, Cell Signaling Technology, Danvers, MA, USA, 1:1000), eukaryotic elongation factor 2 (cat. no. 2332, Cell Signaling Technology, 1:500), S6K1 (cat. no. 9202, Cell Signaling Technology, 1:1000), LAT1 (sc-374232, Santa Cruz Biotechnology, Dallas, TX, USA, 1:100), Unc-51 like autophagy activating kinase 1 (cat. no. NBP2-66765, Novus Biologicals, Abingdon, UK, 1:500), citrate synthetase (cat. no. ab129095, Abcam, Cambridge, UK, 1:1000), complex I (cat. no. ab110242, Abcam, 1:10,000) and complex IV (cat. no. 55070-1-AP, Proteintech, Manchester, UK, 1:10,000). Results are presented in Fig. 9.

To illustrate how the superior speed of THRIFTY can be utilized for novel applications, we also performed high-resolution mitochondrial respirometry on pools of

type I and type II fibres dissected from freshly collected muscle. A muscle biopsy was collected from the vastus lateralis of a healthy 24-year-old female participant from another ongoing study (DNR: 2017/2034-31/2). After collection, the biopsy was immediately placed in ice-cold preservation medium (BIOPS) and 80 single fibres were dissected out and typed as described previously for wet muscle. A total of 36 type I and 39 type II fibres were pooled for subsequent analysis and the total time for dissection, typing and pooling was less than 6 h.

After pooling, the fibres were permeabilized for 15 min with saponin (50 $\mu$g/ml BIOPS) after which they were washed in BIOPS. Type I and type II pools were then transferred to separate chambers of the high-resolution respirometer (Oxygraph-2k, Oroboros Instruments Corp., Innsbruck, Austria) with 2 ml respiration medium

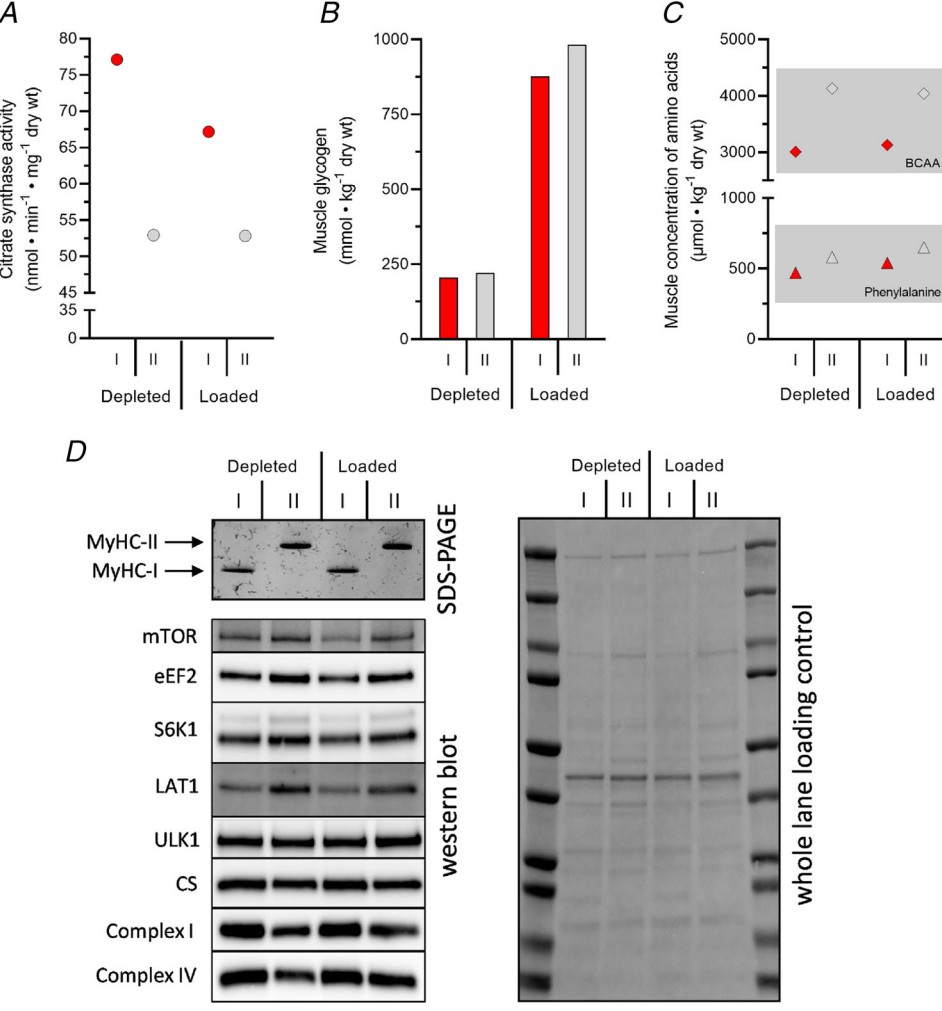

**Figure 9. Application examples of multiple different analyses performed on the same pool of fibres**
Multiple analyses on the same pool of fibres requires a large number of fibres to maximize representativeness of the fibre type population and to yield sufficient sample mass. Here, each pool contained between 83 and 109 fibres, with a sample mass ranging from 423.4 to 644.5 $\mu$g. *A*, muscle citrate synthase activity; *B*, muscle glycogen concentration; *C*, muscle amino acid concentrations (BCAA; sum of leucine, valine and isoleucine); *D*, total levels of several proteins. [Colour figure can be viewed at wileyonlinelibrary.com]

(0.5 mM EGTA, 3 mM $MgCl_2 \cdot 6H_2O$, 60 mM potassium lactobionate, 20 mM taurine, 10 mM $KH_2PO_4$, 20 mM HEPES, 110 mM sucrose and 1 g/l BSA) in each chamber. Maximal respiration was then measured simultaneously in both chambers at 37°C following the addition of substrates (2.5 mM ADP, 0.2 mM octanoyl carnitine, 0.5 mM malate, 10 mM glutamate, 5 mM pyruvate and 10 mM succinate). Outer mitochondrial membrane-integrity was assessed by subsequent addition of cytochrome $c$ (10 $\mu$M). A minor increase in respiration (9%) indicated that the outer membranes were intact. Prior to analysis, $O_2$ calibration was performed according to the manufacturer's instructions. Data acquisition was performed using DatLab 5.2 software (Oroboros, Paar, Graz, Austria). Results are presented in Fig. 10.

## Discussion

Much insight in fibre type-specific physiology has been obtained over the past few decades from studies performed on individually dissected muscle fibres. However, due to the extreme time requirements associated with dissection and fibre typing, many studies have included relatively few fibres. For the same reason, dissection of fibres is often performed on a low number of biopsy specimens from relatively few participants. As such, poor representativeness of the fibre type population, as well as the study population, is often mentioned as a limitation to studies on isolated muscle fibres (for a recent review see Tobias & Galpin, 2020).

While the exact number will undoubtedly vary depending on the specific variable, there is very limited information about the minimal number of fibres required

for an accurate representation of the fibre population. Murach et al. (2016) recently showed that determining fibre type composition on 25 individual fibres gave similar results to when determining it on 125 fibres; however, the authors noted an increased variation when using only 25 fibres. Christiansen et al. (2019) performed mathematic simulation on their sample size of 40 fibres and found that 3–9 fibres were representative with regard to total protein expression of several proteins. However, as these variables are relatively static, i.e. do not change rapidly, the above findings are not that surprising.

In contrast, examining variables with faster turnover and broader dynamic range may require substantially more fibres due to significant variation between fibres of the same type. For instance, Essen & Henriksson (1974) demonstrated a 7- to 12-fold difference in glycogen content between fibres of the same type at rest as well as after exercise. Large variations have also been reported for ATP concentration ($\sim$4–6-fold) (Soderlund & Hultman, 1990), PCr (2–3-fold) (Sahlin et al., 1997) and NADH ($\sim$50%) (Ren et al., 1988). Also, our laboratory recently demonstrated an 80-fold variation in protein phosphorylation between fibres of the same type in response to a combined exercise and nutritional stimulus (Edman et al., 2019). Given these large variations, pooling too few fibres may prevent proper identification of mechanistic links between different variables. This may especially be true if different pools of fibres are generated for different analyses. Thus, ideally, all analyses should be performed on the same pool of fibres from a sample, and to provide an accurate representation of the fibre population and allow for proper resolution and accuracy of certain measurements, the pool should consist of many fibres. However, the exact number of fibres will depend on the specific variables to be measured. To increase the feasibility of performing large scale fibre type-specific analyses, we have developed THRIFTY, a valid high-throughput method for efficient fibre type identification.

To validate THRIFTY, we used dot blot as a reference method (Christiansen et al., 2019), as we considered it to be the most practical approach for fibre typing a large number of samples (400 samples). In this regard, THRIFTY displayed excellent validity in comparison to dot blot ($\kappa = 0.955$, $P < 0.001$), but it must be noted that 9 out of the 400 fibres were typed differently by the two methods. Importantly, however, there was not a single case where one sample was considered pure with THRIFTY for one fibre type (e.g. type I), whilst being considered pure for the opposite fibre type using dot blot (e.g. type II). Thus, the few cases of inconclusive typing occurred on fibres considered hybrid fibres or dual segments using THRIFTY or fibres considered contaminated using dot blot. As the main premise in developing THRIFTY was to facilitate the generation of larger pools of fibres to

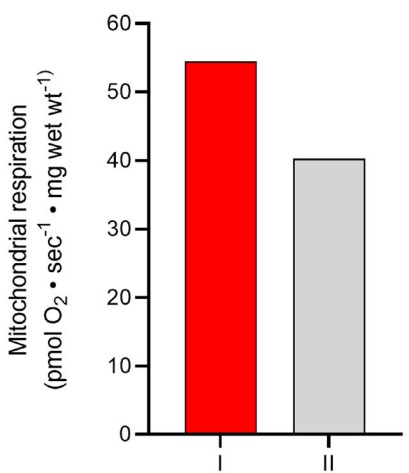

**Figure 10. Maximal mitochondrial respiration measured in pools of type I ($n = 36$) and type II ($n = 39$) fibres from one individual**
Dissection, typing and pooling of fibres was completed under 6 h. [Colour figure can be viewed at wileyonlinelibrary.com]

increase the representativeness of the fibre type population and generate enough sample mass for multiple analyses, a secondary validation of THRIFTY was performed under such conditions. Fibres of the same type, classified with THRIFTY as well as dot blot, were pooled together and the purity of each pool was assessed by SDS-PAGE and silver staining. This analysis revealed that fibre pools generated by both methods were completely pure, lending further support to the validity of THRIFTY.

In addition to high validity, another advantage of THRIFTY is the ability to more reliably identify hybrid fibres, which are characterized by the presence of two or more MyHC isoforms (Medler, 2019). To date, sfSDS-PAGE is considered the 'gold standard' for resolving hybrid fibres as it allows for relative quantification of the MyHC isoforms (Tobias & Galpin, 2020). However, with this method, it is difficult to fully determine if the presence of multiple MyHC isoforms represents a bona fide hybrid fibre or a contamination remaining from the dissection process (see example in Fig. 3) when working with non-skinned fibres. As the sample is denatured prior to gel separation, it must be assumed that all individual fibres were perfectly dissected with no remnants of other fibres. While this is a minor issue in the hands of a skilled operator, there is no guarantee as minor contaminations may be difficult to detect during dissection (see example in Fig. 3). Consequently, the failure to distinguish between contaminated samples and true hybrids may result in the overestimation of the hybrid fibre population. This is also an issue with the dot blot method and conventional single fibre western blot as they rely on the same denaturing sample preparation process. In contrast, owing to the unique design of THRIFTY it is possible to visually identify hybrid fibres based on the antibody binding pattern, thereby excluding any form of contamination. This way of identifying hybrid fibres represents an advantage over the methods mentioned above. Furthermore, for samples with relatively few hybrid fibres, THRIFTY is the most practical and time-saving approach for scanning and eventually identifying a sufficient number of these fibres.

A high-throughput method must not only be valid, but also rapid. Thus, once we had confirmed the validity of THRIFTY, we performed a detailed assessment of the time requirements and compared those to that of the dot blot method. Owing to the lengthy separation times required for sfSDS-PAGE, and the added need for protein transfer in sfWB, these are usually considered to be low throughput methods. In contrast, the recently developed dot blot method presented by Christiansen et al. (2019), offered significant improvements in speed by circumventing these steps, and the authors concluded that it was suitable for typing hundreds of fibres per day. However, when validating their method, they only used 40 individual fibres and provided no detailed rationale

for this conclusion. It is therefore difficult to fully assess to what extent the dot blot could be considered a high throughput method. Here, we made systematic, side-by-side time comparisons for all steps required for the typing of 400 fibres with both THRIFTY and dot blot. We considered this to be an important aspect, partly as this has not been addressed previously, but primarily because the two methods include substantially different steps that should be considered for an accurate comparison. Accounting for all steps, we found that THRIFTY was more than three times faster than the dot blot method ($\sim$11 h *vs.* 37 h, respectively). Given that studies on isolated muscle fibres are limited mainly by their time-consuming nature, this should be viewed as a major advantage. However, some aspects of this comparison warrant further elaboration. First, the timing experiment was performed by an operator novice to both methods. The reason for this design choice was to exclude skill as a determining factor of the outcome, thereby providing a minimally biased measure of the time requirement for each method, and thus the fairest comparison. However, in practice, skill will undoubtedly be relevant for the speed and practicality of most methods and we therefore fully acknowledge that the time requirement of 37 h for the dot blot method may be viewed as excessively long. This is primarily accounted for by the staining procedure, which was the most time-consuming step. To minimize the risk of human error in the hands of a novice operator, the 16 membranes were stained four at a time, with each round taking approximately 4.5 h, totalling approximately 18 h. We do recognize, however, that an experienced operator would be able to handle substantially more membranes simultaneously, perhaps even all 16 at once, which would increase efficiency considerably. Yet, even with a four-fold time reduction of the dot blot staining procedure, the major advantage of THRIFTY still remains with only half the time required from start to finish (i.e. $\sim$11 h *vs.* 24 h). Secondly, it must be noted that the dot blot method, as employed here, deviates from the original outline in that only a small segment was cut off each fibre and subsequently placed in tubes, instead of the whole fibre. This modification obviously lengthened the typing protocol but was an absolute requisite to make the method applicable for generating pools of fibres for downstream analyses with multiple and diverse methods (i.e. not just western blot), and thus for a relevant comparison to THRIFTY.

Beyond the significant improvement in fibre typing efficiency, an added benefit of THRIFTY is the excellent scalability of the staining procedure. The minimalistic design format allows for multiple slides to be stained in parallel, with very little additional time required per added slide. Thus, the time required to stain 1000 segments is virtually the same as for 200 segments. The wide scalability of THRIFTY therefore significantly reduces the required

staining time per fibre. This is in stark contrast to the dot blot method in which a five-fold increase in the number of membranes would result in a substantial increase in accumulated time, and thus an increase in staining time per fibre. Also, as opposed to THRIFTY, where five slides require the same volumes as one, with each added membrane there is an accompanying increase in reagent volume. Considering absolute volumes, simultaneous staining of five slides with 1000 fibre segments using THRIFTY requires only 90 ml of reagents (10 ml acetone, 10 ml of each antibody solution and 60 ml PBS), which is equivalent to the volume required for a single membrane. As staining of 1000 fibre segments would require 37 membranes with dot blot, THRIFTY requires 37 times less antibody and washing solution. THRIFTY also does not require the use of materials and reagents such as Eppendorf tubes, sample buffer, chemiluminescent substrate and PVDF membranes. As such, THRIFTY is significantly less resource demanding and much less costly, even when considering the added expense of the slides and the printing of the grid on each slide (~0.8 €/slide/200 segments).

Many of the advantages of THRIFTY may be attributed to one key factor, namely the printing of a customized grid system on standard microscope slides (Fig. 2). The size of each square within the grid was designed to perfectly fit the field of view of the ×4 objective of our microscope, making it very easy to move from square to square during the typing process. Seamless navigation across the slide was further facilitated by the white colour of the grid as it reabsorbed some of the fluorescence signal emitted in the EYFP filter. This feature bypasses the need to change image settings repeatedly and provides a future potential to automatize the typing process. Importantly, by modifying the printing template, the grid system can easily be altered to include a larger number of squares for even higher throughput, and each square can easily be adjusted to fit the field of view of any microscopy system.

To make the many advantages of THRIFTY available for the wider muscle research community, we also determined if the method could be modified to be used with a standard western blot imaging system equipped with multiplexing capabilities (ChemiDoc MP). With minor modifications to the original protocol, we found that type I and type II fibre pools generated with the modified version of THRIFTY displayed absolute purity without any signs of contamination as assessed by SDS-PAGE and silver staining (Fig. 8). For laboratories with limited access to a fluorescence microscope, this represents a major benefit. However, there are some points that require consideration. First, regardless of the secondary antibody used, there was noticeable background fluorescence in all fibre segments, irrespective of the fluorescence channel, which resulted in considerably lower signal-to-noise ratio compared to the fluorescence microscope. Thus, for accurate fibre type identification using the modified version of THRIFTY, it is absolutely necessary to compare the signal intensity between the two channels. However, for this comparison to be meaningful, one must first create a reference slide with segments of known fibre types to determine the range of signal intensity in each channel, and thus the range of the relative difference in signal intensity between channels. Here we used the original THRIFTY protocol for typing the fibres to be used for the reference slide, but any fibre typing method can be used for this purpose. Thus, any laboratory looking to transition into using the modified THRIFTY protocol can do so with minimal effort. Secondly, due to the lower resolution and limited magnification capabilities of the western blot system, closer inspection of fibre segments is not possible. Consequently, a larger number of fibres are at risk of being designated as inconclusive and therefore excluded to maintain fibre pool purity. For the same reason, accurately identifying hybrid fibres is not feasible. Nevertheless, for laboratories with limited access to a fluorescence microscope, the modified THRIFTY method remains highly useful due to its vast improvements in speed and practicality.

A potential limitation of THRIFTY, as presented here, may be the lack of differentiation between type IIA and type IIX subclasses. We chose not to differentiate between the two primarily based on previous reports showing very low abundance of pure type IIX fibres in healthy individuals (Fry et al., 2014; Horwath et al., 2021; Murach et al., 2019). Given the low abundance samples available to us, dissecting and identifying a large enough number of type IIX fibres for validation purposes was beyond the scope of this proof-of-concept study. We would furthermore argue that performing studies on large pools of pure type IIX fibres would be of much less relevance in a healthy and active population. Nevertheless, it is acknowledged that substantial proportions of type IIX fibres have been reported in states of muscle disuse (Andersen et al., 1996; Vikne et al., 2020), and that validated identification of type IIX fibres using THRIFTY would be beneficial for laboratories with such a research focus. However, as fibre typing with THRIFTY is based on subclass-specific antibody binding, for laboratories with a particular interest in type IIX fibres and access to antibodies recognizing this subclass, the protocol can easily be modified and validated in-house.

Beyond the many benefits described above, the superior speed of THRIFTY can also be taken advantage of for time-sensitive applications. One such application is the measurement of mitochondrial respiration, which must be performed within hours of tissue collection. Mitochondrial respiration remains stable up to 6 h after sampling (Cardinale et al., 2018) but is significantly reduced after 24 h in cold preservation medium (Gnaiger et al., 2000). Given the time-sensitive nature

of mitochondrial respiration, and the lengthy staining protocols of other contemporary methods, not including the fixed time of fibre dissection, fibre type-specific mitochondrial respiration has not been possible to measure until now. With THRIFTY's rapid staining protocol of less than 2 h, sufficient time remains for dissecting and pooling enough fibres to measure fibre type-specific respiration (see Fig. 10). Thus, THRIFTY does not only increase the feasibility to conduct larger fibre type-specific studies, but also provides new opportunities to gain novel insights into human fibre type-specific metabolism.

In conclusion, THRIFTY is a novel high-throughput method for fibre type identification of isolated muscle fibres. THRIFTY has excellent validity in comparison to existing methods but is considerably less resource demanding. Due to its unique design features, THRIFTY has excellent scalability and allows for fibre typing at superior speeds, even in novice hands, and as such, THRIFTY encompasses all the hallmarks of a true high-throughput method. Of special note is the fact that with some experience, a skilled operator can utilize THRIFTY at twice the speed of a novice. Thus, an experienced operator can type up to 800 individually dissected muscle fibres in just under 11 h. Beyond increasing the feasibility for larger scale fibre type-specific studies, the superior speed of THRIFTY can also be used to perform time-sensitive assays such as high-resolution mitochondrial respirometry where measurements need to be carried out in close connection with tissue sampling. A final advantage of THRIFTY is its high versatility, which allows for simple implementation in laboratories with access to either a fluorescence microscope or western blot imaging system with fluorescence multiplexing capabilities. We therefore consider THRIFTY an important tool for future work and we believe its implementation will provide new opportunities to gain further insights into fibre type-specific muscle physiology.

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

## Additional information

### Data availability statement

The data that support the findings of this study are available from the corresponding author upon reasonable request.

## Competing interests

None.

## Author contributions

O.H. and W.A. conceived the idea behind this work; O.H., S.E., A.A., F.J.L. and W.A. acquired and interpreted data; S.E. performed statistical analysis; O.H. drafted the manuscript; S.E., A.A., F.J.L. provided critical feedback and W.A. finalized the manuscript. All authors have read and approved the final version of this manuscript and agree to be accountable for all aspects of the work in ensuring that questions related to the accuracy or integrity of any part of the work are appropriately investigated and resolved. All persons designated as authors qualify for authorship, and all those who qualify for authorship are listed.

## Funding

This study was funded by project grants (P2020-0058, P2021-0173) awarded to W.A. from the Swedish Research Council for Sport Science. During this work W.A. was also supported by an Early Career Research Fellowship from the Swedish National Centre for Research in Sports (no. D2019-0050).

## Acknowledgements

The BA-F8 and SC-71 antibodies, developed by S. Schiaffino from University of Padova, were obtained from the Developmental Studies Hybridoma Bank, created by the NICHD of the NIH and maintained at The University of Iowa, Department of Biology, Iowa City, IA 52242, USA. The authors are grateful to Dr Marcus Moberg for technical support and for providing constructive feedback on this manuscript.

## Keywords

fibre typing, high-throughput, immunofluorescence, MyHC, SDS-PAGE

## Supporting information

Additional supporting information can be found online in the Supporting Information section at the end of the HTML view of the article. Supporting information files available:

**Statistical Summary Document**
**Peer Review History**

