## [Peer Review History · The Journal of Physiology]

THRIFTY, a novel high-throughput method for rapid fibre type identification of isolated skeletal muscle fibres

Oscar Horwath, Sebastian Edman, Alva Andersson, Filip J Larsen, and William Apró
DOI: 10.1113/JP282959

Corresponding author(s): William Apró (william.apro@gih.se)

Review Timeline:

Submission Date:	08-Feb-2022
Editorial Decision:	11-Mar-2022
Resubmission Received:	25-Apr-2022
Editorial Decision:	22-Jun-2022
Revision Received:	22-Jul-2022
Accepted:	25-Aug-2022

Senior Editor: Scott Powers

Reviewing Editor: Michael Hogan

Transaction Report:

Dear Dr Apró,

Re: JP-TFP-2022-282959 "THRIFTY, a novel high-throughput method for rapid fibre type identification of isolated skeletal muscle fibres" by Oscar Horwath, Sebastian Edman, Alva Andersso, and William Apró

Thank you for submitting your manuscript to The Journal of Physiology. It has been assessed by a Reviewing Editor and by 2 Referees and the reports are copied below.

I regret to say that the manuscript has not been accepted for publication.

Some positive comments were made on the manuscript. Unfortunately, they did not outweigh the more serious criticisms which led the Reviewing Editor to recommend rejection.

I am sorry to have to pass on this disappointing news, and hope it will not discourage you from making future submissions of new work to The Journal of Physiology.

However, we believe your manuscript is worthy of further consideration and suggest that you transfer your manuscript to Physiological Reports (<https://physoc.onlinelibrary.wiley.com/hub/journal/2051817X/aims-and-scope/read-full-aims-and-scope>), a peer-reviewed, open access, interdisciplinary journal, jointly owned by the American Physiological Society and The Physiological Society.

To transfer your manuscript to Physiological Reports, the corresponding author must send authorization within 120 days of receipt of this letter. Please use this link Transfer to Physiological Reports to send an authorization email to transfer your manuscript. If your manuscript does not require additional peer review, the editors of Physiological Reports will aim to give you an initial decision within 3 working days. In fact, >80% of transferred submissions are accepted for publication. Please note, of course, that we cannot guarantee final acceptance.

I hope you will take advantage of the opportunity to allow the editors of Physiological Reports to evaluate your manuscript.

You may be able to publish Open Access with no direct cost to yourself. You can check your eligibility here <https://secure.wiley.com/openaccess?>

Yours sincerely,

Scott K. Powers
Senior Editor
The Journal of Physiology
<https://jp.msubmit.net>
<http://jp.physoc.org>
The Physiological Society
Hodgkin Huxley House
30 Farringdon Lane
London, EC1R 3AW
UK
<http://www.physoc.org>
<http://journals.physoc.org>

EDITOR COMMENTS

Reviewing Editor:

Your manuscript has now been reviewed by two referees with expertise in the study of isolated skeletal muscle fibers. Overall, it was felt that the studies were well conducted and the importance of developing a high-throughput method for the typing analysis of isolated skeletal muscle fibers was well recognized. However, concerns were raised about whether the THRIFTY technique really overcomes the major barriers that currently exist (e.g., the isolation of the single muscle fibers). There were also concerns that the potential need for specialized equipment such as the CELENA would limit the use of the THRIFTY technique. Collectively, it was concluded that there is not enough evidence to indicate that the technique will significantly improve our ability answer questions of physiological significance. Thus, the potential impact of the work was not rated high enough to warrant further consideration.

Senior Editor:

Thank you for submitting your work to the Journal of Physiology. Your report has been reviewed by two referees and a review editor (RE) who are experts in the field. As noted in the RE comments, the collective decision by the referees and RE is that insufficient evidence exists that the proposed technique will significantly improve our ability answer key physiological

questions involving skeletal muscle. Therefore, the impact of your report was not rated high enough to warrant further review. I hope that this decision will not prevent you from submitting future work to the Journal of Physiology.

REFEREE COMMENTS

Referee #1:

This is a well written paper by Horwath and colleagues that describes a high-throughput method (400 fibers) for skeletal muscle fiber typing (termed THRIFTY) that reduces analysis time by ~70% when compared to dot blotting. The THRIFTY method was validated using SDS-PAGE (gold standard) and dot blotting where the authors saw no differences in fiber type distribution between methodologies. The authors provide a detailed workflow with equipment and infrastructure necessary to apply the THRIFTY approach. The paper focuses on MHC I and IIA muscle fiber types with additional directions how to probe for other proteins targets like MHC IIX that are highly relevant to human muscle fiber typing. Overall, the THRIFTY approach should be useful to a wide range of researchers involved with muscle fiber typing and fiber type specific physiology/biology.

1. The authors highlight that SDS-PAGE is the gold standard and is used in their work to help validate the THRIFTY approach. The comment on page 17 about contamination from the dissection process is a minor concern and seems exaggerated as a reason not to use the gold standard approach. There are various solutions that are viable for isolating muscle fibers (RNAlater, RNA-best, etc.) that make isolation quite easy, reliable, and provide opportunities for various applications (fiber type, fiber pooling, protein identification, molecular probing). In contrast the freeze-dried muscle fiber isolation can be rather labor some and also requires a special skillset. Thus, it would depend upon the research question and approach to which technique might be most applicable.
2. The Murach 2016 paper studied resistance trained individuals (presented on page 16), which given their training status would be expected to have few hybrid (<10%) muscle fibers. However, the amount of hybrid fibers can easily scale to >20% or more with inactivity and prolonged disuse, which may alter the fidelity of small fiber populations to accurately reflect the fiber type profile.
3. Supplemental Figures 2 & 3 are not totally visible and appear to be cropped off the page.
4. Cost is mentioned for the slides and printing grid but is a mention of the cost of the CELENA S Digital Imaging System worth it as these microscopes can be a huge expense to consider to be able to do THRIFTY. Something you don't need for SDS-PAGE.
5. The statement on how hybrids are visually determined in THRIFTY (page 18, line 2) would be better placed in the methods section.
6. "Ques" to "cues" on key point 2.
7. Page 5: lowercase "western blot" to keep consistent. Apply to rest of paper.
8. Page 11: To give context to inconclusive THRIFTY hybrid and dual fibre segments (Seven fibre segments were deemed inconclusive.....) consider referring the reader to Figure 3 for a visual example.
9. Page 12: Change "se" to "see" on 3rd line of last paragraph.
10. Page 16: missing parenthesis at the end first discussion paragraph.
11. Page 28: Supplemental Figure 3 legend is incorrectly labeled "Figure S2."

Referee #2:

The authors describe a proposed new method (with the acronym THRIFTY) for a more rapid identification of MyHC isoform content (type 1 vs type II) in single fiber segments from human biopsy samples. The technique involves the dissection of single fiber segments from a biopsy samples, the arrangement of these individual fiber segments onto a grid on a microscope slide and then using currently existing and well used techniques (primary AB to MyHC-I and MyHC-II, followed by fluorescently labelled second Ab and use of a digital fluorescent microscope) to visualize these fibers and their MyHC isoform content.

While the technique is well described and is somewhat validated against some other existing techniques, I have the following concerns:

1 - It appears that the only really novel contribution of this technique is the grid pattern printed onto the microscope slide. All other aspects of this technique are well developed and highly used currently in the field.

2- The authors correctly point out that identification of fiber type (or MyHC isoform phenotype) is a painstaking process that often limits the number of fibers that can be tested. However, it is not obvious to this reviewer how this new technique significantly alters the painstaking nature of this process. The limiting factor of this technique remains the dissection of hundreds of individual fibers from biopsy samples. Even though this proposed technique may involve simpler or faster steps after the dissection - compared to dot blot or SDS-PAGE - the need to individually dissect fibers from the sample keeps this technique from being a true "high-throughput technique". Several published papers have done MyHC isoform analysis of more than 1000 fibers using SDS-PAGE analysis.

3- The paper would strongly benefit from some examples of how this technique might be used to better answer questions of physiological significance compared to other existing techniques of single fiber identification. Even better would be for this technique to be used *within this paper* to answer a physiological question, in order to show the power of this technique. For example, if the question to be answered is simply how does a given intervention or condition affect fiber type (as given by MyHC isoform) distribution in a muscle, this is much more quickly and completely answered using immunohistochemistry on cross-sections of muscle. With IHC on cross-sections one can visualize MyHC isoform content of all fibers in a cross-section immediately with no need for painstaking fiber dissection. If the point is for the THIRTY method to be able to connect the MyHC isoform content of a fiber to some other measurement in that fiber (contractile function, etc.) then it is likely that this other measurement would be the limiting factor in throughput and therefore the small amount of time gained with the THIRTY method would be inconsequential in improving throughput. Without seeing the THIRTY method actually used to answer a physiological question it is difficult to gauge its importance.

4- The authors compare the THIRTY results to SDS-PAGE analysis of MyHC isoform content as a way of validating the THIRTY technique, but make the curious choice to pool all samples identified as a particular type and run those on a SDS-PAGE gel. The true power of the SDS-PAGE analysis of single fiber MyHC isoform content is the ability to determine MyHC isoforms in individual fibers. The proper validity comparison is to compare the SDS-PAGE results from a large number of individual fiber segments with the results given by the THIRTY analysis of those same fibers, thus showing that the THIRTY method has the same level of accuracy as SDS-PAGE for on an individual fiber identification.

Dear Dr Apró,

Re: JP-TFP-2022-283262X "THRIFTY, a novel high-throughput method for rapid fibre type identification of isolated skeletal muscle fibres" by Oscar Horwath, Sebastian Edman, Alva Andersson, and William Apró

Your appeal to the recent decision on your Journal of Physiology manuscript has been carefully considered by both the review editor and the senior editor. I am pleased to inform you that your appeal has been accepted and therefore, I would like to provide you with one final opportunity to revise your report for further review. Please note that this acceptance of your appeal does NOT guarantee acceptance of the revised paper. Indeed, a final decision about acceptance will be made after re-review and further evaluation by the review editor. Thank you for your patience with this process. We look forward to receiving your revised report.

The reports are copied at the end of this email. Please address all of the points and incorporate all requested revisions, or explain in your Response to Referees why a change has not been made.

NEW POLICY: In order to improve the transparency of its peer review process The Journal of Physiology publishes online as supporting information the peer review history of all articles accepted for publication. Readers will have access to decision letters, including all Editors' comments and referee reports, for each version of the manuscript and any author responses to peer review comments. Referees can decide whether or not they wish to be named on the peer review history document.

Authors are asked to use The Journal's premium BioRender (<https://biorender.com/>) account to create/redrawn their Abstract Figures. Information on how to access The Journal's premium BioRender account is here: <https://physoc.onlinelibrary.wiley.com/journal/14697793/biorender-access> and authors are expected to use this service. This will enable Authors to download high-resolution versions of their figures.

I hope you will find the comments helpful and have no difficulty returning revisions within 4 weeks.

If you need to check to make sure that your Methods section conforms to the principles of UK regulations, you may wish to refer to Grundy (2015):

Grundy (2015) J. Physiol. 2015 Jun 15;593(12):2547-9 <https://doi.org/10.1113/JP270818>

Your revised manuscript should be submitted online using the links in Author Tasks Link Not Available. This link is to the Corresponding Author's own account, if this will cause any problems when submitting the revised version please contact us.

The image files from the previous version are retained on the system. Please ensure you replace or remove any files that have been revised.

REVISION CHECKLIST:

- Summary data must be reported as mean {plus minus} SD or 95% confidence interval
- All table and figure legends with summary data must include the statistical test used in the table/figure and sample size
- Figures with summary data bars must include individual data points, or box whisker plots when $n < 30$.
- Article file, including any tables and figure legends, must be in an editable format (eg Word)
- Abstract figure file (see above)
- Statistical Summary Document
- Upload each figure as a separate high quality file
- Upload a full Response to Referees, including a response to any Senior and Reviewing Editor Comments;
- Upload a copy of the manuscript with the changes highlighted.

- A potential 'Cover Art' file for consideration as the Issue's cover image;

- Appropriate Supporting Information (Video, audio or data set https://jp.msubmit.net/cgi-bin/main.plex?form_type=display_requirements#supp).

To create your 'Response to Referees' copy all the reports, including any comments from the Senior and Reviewing Editors, into a Word, or similar, file and respond to each point in colour or CAPITALS and upload this when you submit your revision.

I look forward to receiving your revised submission.

If you have any queries please reply to this email and staff will be pleased to assist.

Yours sincerely,

Scott K. Powers
Senior Editor
The Journal of Physiology
<https://jp.msubmit.net>
<http://jp.physoc.org>
The Physiological Society
Hodgkin Huxley House
30 Farringdon Lane
London, EC1R 3AW
UK
<http://www.physoc.org>
<http://journals.physoc.org>

REQUIRED ITEMS:

-You must start the Methods section with a paragraph headed Ethical Approval. If experiments were conducted on humans confirmation that informed consent was obtained, preferably in writing, that the studies conformed to the standards set by the latest revision of the Declaration of Helsinki, and that the procedures were approved by a properly constituted ethics committee, which should be named, must be included in the article file. If the research study was registered (clause 35 of the Declaration of Helsinki) the registration database should be indicated, otherwise the lack of registration should be noted as an exception (e.g. The study conformed to the standards set by the Declaration of Helsinki, except for registration in a database.). For further information see: <https://physoc.onlinelibrary.wiley.com/hub/human-experiments>

-The Reference List must be in Journal format

-Your manuscript must include a complete Additional Information section

-You must upload original, uncropped western blot/gel images (including controls) if they are not included in the manuscript. This is to confirm that no inappropriate, unethical or misleading image manipulation has occurred <https://physoc.onlinelibrary.wiley.com/hub/journal-policies#imagmanip> These should be uploaded as 'Supporting information for review process only'. Please label/highlight the original gels so that we can clearly see which sections/lanes have been used in the manuscript figures.

-Your paper contains Supporting Information of a type that we no longer publish. Any information essential to an understanding of the paper must be included as part of the main manuscript and figures. The only Supporting Information that we publish are video and audio, 3D structures, program codes and large data files. Your revised paper will be returned to you if it does not adhere to our Supporting Information Guidelines

-A Statistical Summary Document, summarising the statistics presented in the manuscript, is required upon revision. It must be on the Journal's template, which can be downloaded from the link in the Statistical Summary Document section here: https://jp.msubmit.net/cgi-bin/main.plex?form_type=display_requirements#statistics

-Papers must comply with the Statistics Policy https://jp.msubmit.net/cgi-bin/main.plex?form_type=display_requirements#statistics

In summary:

-If n {less than or equal to} 30, all data points must be plotted in the figure in a way that reveals their range and distribution. A bar graph with data points overlaid, a box and whisker plot or a violin plot (preferably with data points included) are acceptable formats.

-If $n > 30$, then the entire raw dataset must be made available either as supporting information, or hosted on a not-for-profit repository e.g. FigShare, with access details provided in the manuscript.

-' n ' clearly defined (e.g. x cells from y slices in z animals) in the Methods. Authors should be mindful of pseudoreplication.

-All relevant ' n ' values must be clearly stated in the main text, figures and tables, and the Statistical Summary Document (required upon revision)

-The most appropriate summary statistic (e.g. mean or median and standard deviation) must be used. Standard Error of the Mean (SEM) alone is not permitted.

-Exact p values must be stated. Authors must not use 'greater than' or 'less than'. Exact p values must be stated to three significant figures even when 'no statistical significance' is claimed.

-Statistics Summary Document completed appropriately upon revision

-A Data Availability Statement is required for all papers reporting original data. This must be in the Additional Information section of the manuscript itself. It must have the paragraph heading "Data Availability Statement". All data supporting the results in the paper must be either: in the paper itself; uploaded as Supporting Information for Online Publication; or archived in an appropriate public repository. The statement needs to describe the availability or the absence of shared data. Authors must include in their Statement: a link to the repository they have used, or a statement that it is available as Supporting Information; reference the data in the appropriate sections(s) of their manuscript; and cite the data they have shared in the References section. Whenever possible the scripts and other artefacts used to generate the analyses presented in the paper should also be publicly archived. If sharing data compromises ethical standards or legal requirements then authors are not expected to share it, but must note this in their Statement. For more information, see our Statistics Policy.

-Please include an Abstract Figure. The Abstract Figure is a piece of artwork designed to give readers an immediate understanding of the research and should summarise the main conclusions. If possible, the image should be easily 'readable' from left to right or top to bottom. It should show the physiological relevance of the manuscript so readers can assess the importance and content of its findings. Abstract Figures should not merely recapitulate other figures in the manuscript. Please try to keep the diagram as simple as possible and without superfluous information that may distract from the main conclusion(s). Abstract Figures must be provided by authors no later than the revised manuscript stage and should be uploaded as a separate file during online submission labelled as File Type 'Abstract Figure'. Please ensure that you include the figure legend in the main article file. All Abstract Figures should be created using BioRender. Authors should use The Journal's premium BioRender account to export high-resolution images. Details on how to use and access the premium account are included as part of this email.

-Author photo and profile. First (or joint first) authors are asked to provide a short biography (no more than 100 words for one author or 150 words in total for joint first authors) and a portrait photograph. These should be uploaded and clearly labelled with the revised version of the manuscript. See Information for Authors for further details.

EDITOR COMMENTS

Reviewing Editor:

While the 2 new reviewers of this manuscript felt that your study has significant merit, both reviewers felt that more experiments would be necessary to push this study into being of great interest to the readership of the Journal of

Physiology. We are sorry that this has been an extended review process, but the 2 new reviewers did not provide any additional support for further consideration of your study.

Senior Editor:

Your appeal to the recent decision on your Journal of Physiology manuscript has been carefully considered by both the review editor and the senior editor. I am pleased to inform you that your appeal has been accepted and therefore, I would like to provide you with one final opportunity to revise your report for further review. Please note that this acceptance of your appeal does NOT guarantee acceptance of the revised paper. Indeed, a final decision about acceptance will be made after re-review and further evaluation by the review editor. Thank you for your patience with this process. We look forward to receiving your revised report.

No statistics summary has been submitted.

REFEREE COMMENTS

Referee #1:

This is a revised (R1) paper that is improved over the first submission. The additions to the manuscript add clarity and also provide a better scientific foundation for the THRIFTY approach. The newly included "Applications" sections highlights in better detail the ability of the THRIFTY protocol to run specific assays at the fibre type level (i.e. metabolites, mass spec, PCR, proteomics, etc.). Additionally, Figure 4 and Supplement Figure 4 provide greater detail into the THRIFTY protocols fiber type specific applications and some data from the method.

The muscle glycogen content shown in slow and fast muscle fibers is on the high end (1000 mmol/kg dry wt) compared to previous literature in this area (Essen et al., *Acta Physiol Scand* 95:153-165, 1975 and Greenhaff et al., *J Physiol*, 478: 149-155, 1984). Please comment.

Referee #3:

In this manuscript, the authors describe an improved protocol for rapid fiber the determination on isolated muscle fiber segments. Overall, the work was conducted carefully and the article is well-written. I appreciate the effort put into this work and think the method has the potential to be adopted by the skeletal muscle field.

I think the authors could do a better job of describing how most aspects of fiber typing single fibers is a "fixed cost" from a time perspective. For instance, isolating individual fibers is tedious no matter who is doing it; for even for the most skilled technician, this is going to take time (i.e. is more-or-less fixed cost). Where THRIFTY obviously shines is the actual fiber-typing. Where single fiber SDS-PAGE may take 18-24 hours, the THRIFTY technique only takes a few hours. This time savings opens up all kinds of opportunities that weren't fully explored in this manuscript (which the authors highlight in the last paragraph). I think this point could also be reinforced in the Key Points at the beginning of the manuscript.

In order to fully realize the potential of the THRIFTY technique, the authors should perform respirometry on "wet" slow and fast muscle fibers from humans. The THRIFTY technique will make this possible since those types of experiments must be performed "fresh", or within a matter of hours from tissue collection. This could be an n=1 proof of concept experiment, but would drive home how impactful this technique could be for the field, as well as provide some novel information on fiber type-specific metabolism in humans.

Minor comments:

1. Contamination for sfSDS-PAGE only really occurs when isolating fibers from non-skinned bundles. Granted, that is what is being done here, but I think it is worth differentiating the two.
2. What was the rationale for the antibody dilutions? Also, why use supernatant for the myosin antibodies when concentrate gives you greater control over the amount of antibody being applied?
3. Is a "mailer" analogous to a Coplin jar? Perhaps make that clear.

In Figure 4, I would put a total time estimate somewhere on the Figure. It also seems like this figure could serve as the graphical abstract, perhaps with some additional information on what make the technique advantageous over others?

Referee #4:

The THRIFTY method presented adds some advantage to current published methodologies. The most impressive is that fibres collected are not required to be denatured and so they may be used for measurements other than western blotting. This is a welcomed addition. There are many factors, however, that do not work in the favor of THRIFTY. For instance, the time required for collecting the high number of fibres is no different and is the most consuming step present. As acknowledged in the manuscript, technician experience will dictate the length of this (and other) steps. Other steps may be quicker, but what about the trade-off which is that the single fibres are left at room temperature for a long period whilst fibre typing is taking place. This is particularly problematic if they are not kept in a (denaturing) solution because there may be proteolytic activity occurring during this time. Freeze dried fiber segments are small and may absorb water from the air, which could initiate these events. Whilst the lack of requiring labelling tubes saves time, it may introduce a confounding factor. This would need to be addressed.

Assumably if 1000 fibres are collected to be analysed together, for the cost-time benefit outlined, what validation has been done to ensure that the fibre collection time does not affect downstream steps and measurements?

One of the issues in single fiber methodology is the lack of being able to simply and accurately detect MHC IIx fibers in humans. In this manuscript, this issue remains, however as pointed out by the authors, in muscle samples from healthy individuals this number would be low and so remain out of scope. For a similar reason, the number and type of hybrid fibers that are present, whilst with THRIFTY can be observed (assuming the microscopy method is used), yet once again the number would be low and so downstream measurements would be unlikely. As such, there is no benefit of THRIFTY to add these missing steps for single fiber analyses.

Page 17: "Given these large variations, pooling too few fibres may prevent proper identification of mechanistic links between different variables." And "..... the pool should consist of as many fibres as possible."

Variation is an issue, however a number of fibers should be provided, not 'as many as possible'. One could go on collecting fibers for a very long time, so the number that 'is possible' is very high. Choosing '100' is random, some analyses should be done to arrive at the number of fibres required.

Page 23: "high resolution mitochondrial respirometry" would require fresh tissue, how could the fibres be collected, fiber typed and then used for measurements? This would still be a delay of some hours. In order to make this statement, data should be provided.

Fig S4: appears to show data for a single set of type I or type II fibers. This is not representative, and would need a higher 'n' for each measure and statistics to be included.

END OF COMMENTS

1st Confidential Review

25-Apr-2022

EDITOR COMMENTS

Reviewing Editor:

While the 2 new reviewers of this manuscript felt that your study has significant merit, both reviewers felt that more experiments would be necessary to push this study into being of great interest to the readership of the Journal of Physiology. We are sorry that this has been an extended review process, but the 2 new reviewers did not provide any additional support for further consideration of your study.

Senior Editor:

Your appeal to the recent decision on your Journal of Physiology manuscript has been carefully considered by both the review editor and the senior editor. I am pleased to inform you that your appeal has been accepted and therefore, I would like to provide you with one final opportunity to revise your report for further review. Please note that this acceptance of your appeal does NOT guarantee acceptance of the revised paper. Indeed, a final decision about acceptance will be made after re-review and further evaluation by the review editor. Thank you for your patience with this process. We look forward to receiving your revised report.

No statistics summary has been submitted.

Thank you for the opportunity to submit a revised version of our manuscript. We have performed the requested analyses and we have included a statistics summary.

REFEREE COMMENTS

Referee #1:

This is a revised (R1) paper that is improved over the first submission. The additions to the manuscript add clarity and also provide a better scientific foundation for the THRIFTY approach. The newly included "Applications" sections highlight in better detail the ability of the THRIFTY protocol to run specific assays at the fibre type level (i.e. metabolites, mass spec, PCR, proteomics, etc.). Additionally, Figure 4 and Supplement Figure 4 provide greater detail into the THRIFTY protocols fiber type specific applications and some data from the method.

Thank you for taking the time to once again review our manuscript and for the constructive comments.

The muscle glycogen content shown in slow and fast muscle fibers is on the high end (1000 mmol/kg dry wt) compared to previous literature in this area (Essen et al., Acta Physiol Scand 95:153-165, 1975 and Greenhaff et al., J Physiol, 478: 149-155, 1984). Please comment.

In the ongoing study from which the fibres were dissected we used a protocol consisting of a bilateral glycogen depletion session to maximize subsequent glycogen repletion. This was followed by 2.5 days of carbohydrate loading with 10 g of carbohydrate per kg body weight. Approximately 35% of the carbohydrates were supplied as a glucose polymer with very high molecular mass which has been

shown to enhance glycogen resynthesis (Piehl-Aulin et al. Eur J Appl Physiol. 2000 Mar;81(4):346-51). As such the high values are expected and similar values can be found in the literature. For instance, in the paper by Essen and Henriksson (Acta Physiol Scand. 1974 Mar;90(3):645-7.) the highest reported value is 978 mmol/kg after glycogen depletion and subsequent replenishment, and resting values of 881 mmol/kg are reported by Essén et al. (Acta Physiol Scand 95:153-165, 1975). Similar values (> 900 mmol/kg) have also been reported on whole muscle (Wojtaszewski et al. Am J Physiol Endocrinol Metab. 2003 Apr;284(4):E813-22.).

Referee #3:

In this manuscript, the authors describe an improved protocol for rapid fiber the determination on isolated muscle fiber segments. Overall, the work was conducted carefully and the article is well-written. I appreciate the effort put into this work and think the method has the potential to be adopted by the skeletal muscle field.

Thank you for taking the time to review our manuscript and for providing constructive comments.

I think the authors could do a better job of describing how most aspects of fiber typing single fibers is a "fixed cost" from a time perspective. For instance, isolating individual fibers is tedious no matter who is doing it; for even for the most skilled technician, this is going to take time (i.e. is more-or-less fixed cost). Where THRIFTY obviously shines is the actual fiber-typing. Where single fiber SDS-PAGE may take 18-24 hours, the THRIFTY technique only takes a few hours. This time savings opens up all kinds of opportunities that weren't fully explored in this manuscript (which the authors highlight in the last paragraph). I think this point could also be reinforced in the Key Points at the beginning of the manuscript.

We have now added a more extended description of the fixed time cost in the introduction as well as an expanded discussion about the potential implications of the time savings when using THRIFTY. We have also emphasized this in an additional Key Point.

In order to fully realize the potential of the THRIFTY technique, the authors should perform respirometry on "wet" slow and fast muscle fibers from humans. The THRIFTY technique will make this possible since those types of experiments must be performed "fresh", or within a matter of hours from tissue collection. This could be an n=1 proof of concept experiment, but would drive home how impactful this technique could be for the field, as well as provide some novel information on fiber type-specific metabolism in humans.

We have now performed respirometry on freshly dissected type I and type II from one individual and have included it in the manuscript. Please see figure 10.

Minor comments:

1. Contamination for sfSDS-PAGE only really occurs when isolating fibers from non-skinned bundles. Granted, that is what is being done here, but I think it is worth differentiating the two.

Point taken; this distinction has now been added to the manuscript.

2. What was the rationale for the antibody dilutions? Also, why use supernatant for the myosin antibodies when concentrate gives you greater control over the amount of antibody being applied?

Before the development of THRIFTY, we had been using supernatant for fibre typing muscle cross-sections and therefore had the supernatant antibodies readily available for pilot experiments. Based purely on laboratory practice, pilot experiments were initiated with BA-F8 and SC-71 at 1:1000 but the signal was weaker for SC-71 so the concentration was increased to 1:500 which produced a satisfactory signal.

Based on the information from DSHB, one vial of concentrate contains the same amount of IgG as one vial of supernatant (if produced from the same batch) but in ten times less volume which has been obtained by using concentration filters. It is therefore easy to adjust the volume of the antibody required from ones preferred antibody format. For subsequent typing we have used concentrate instead but at ten-fold lower volumes with no difference in performance.

3. Is a "mailer" analogous to a Coplin jar? Perhaps make that clear.

Yes it is; this has been clarified in the manuscript.

In Figure 4, I would put a total time estimate somewhere on the Figure. It also seems like this figure could serve as the graphical abstract, perhaps with some additional information on what make the technique advantageous over others?

The figure has been changed accordingly. We have also included a graphical abstract with the reviewer's suggestions.

Referee #4:

The THRIFTY method presented adds some advantage to current published methodologies. The most impressive is that fibres collected are not required to be denatured and so they may be used for measurements other than western blotting. This is a welcomed addition.

Thank you for taking the time to review our manuscript and for acknowledging some of the benefits of THRIFTY.

There are many factors, however, that do not work in the favor of THRIFTY. For instance, the time required for collecting the high number of fibres is no different and is the most consuming step present. As acknowledged in the manuscript, technician experience will dictate the length of this (and other) steps.

We fully acknowledge that THRIFTY does not increase the speed of dissection. However, we make no such claim, and the issue of time-consuming dissection remains for all available methods used for single fibre typing. To state that this is a factor specifically not in favor of THRIFTY therefore seems unjustified and unfair.

Skill and experience will always be a factor to consider with all methods. That is why a technician inexperienced in both THRIFTY and dot blot performed all steps for both methods. In our comparison, the lack of experience effectively removes skill as a determining factor, and we show that THRIFTY is three times faster. It is therefore difficult to understand how this can be seen as anything other than a factor in favor of THRIFTY.

Other steps may be quicker, but what about the trade-off which is that the single fibres are left at room temperature for a long period whilst fibre typing is taking place. This is particularly problematic if they are not kept in a (denaturing) solution because there may be proteolytic activity occurring during this time. Freeze dried fiber segments are small and may absorb water from the air, which could initiate these events. Whilst the lack of requiring labelling tubes saves time, it may introduce a confounding factor. This would need to be addressed.

Please note that each individual step in THRIFTY is different from the steps in the dot blot after cutting of the minimal fibre segment to be used for typing, and the number of steps differ between the two methods. One should therefore refrain from trying to compare individual steps and instead look at the performance of the whole method from start to finish. As mentioned before, from start to finish, THRIFTY is considerably quicker.

The reviewer makes a good point regarding the potential trade-off of fibres being in room temperature and the risk of inducing proteolysis due to moisture in the air. It is reasonable to expect that freeze-dried tissue may absorb water from the air due to condensation when moved from the freezer to room temperature. However, the amount of water absorbed is minimal and should have minimal to no effect on biological processes such as proteolysis.

According to Lowry and Passonneau (1972, *A flexible system of enzymatic analysis*, Academic Press, New York), 1% of water is absorbed for each 10% humidity at ambient room temperature. Standard procedure in our laboratory when working with freeze-dried muscle is to maintain humidity below 40% and therefore a maximal increase in fibre weight of less than 4% could be expected due to absorption of water from the air. Equilibration occurs within minutes (Lowry and Passonneau, 1972) after which there should be no further increase in fibre weight. This has been confirmed by Essén et al in their early studies (*Acta Physiol Scand.* 1974 Mar;90(3):645-7 and *Acta Physiol Scand.* 1975, 95:153-165,) in which they weighed fibres repeatedly during the day and saw no increase in fibre weight.

In our laboratory, fibre dissection is always carried out below 40% humidity. We also take additional precautionary measures by always thawing freeze-dried muscle on moisture-absorbing silica-gel for at least 30 minutes prior to dissection. Thawing on silica-gel minimizes condensation which further reduces absorption of water. In our hands, when treated as above, fibre weight increases less than 0.3 % when left out for 24 hours at 21°C and 32% humidity. Given this extremely small increase, there is no reason to expect an increase in proteolysis.

To illustrate this, we compared three experimental conditions.

1: Immediate homogenization after thawing on silica-gel for 30 minutes. Bare minimum exposure to ambient room temperature and low humidity and therefore served as the control condition.

2: Homogenization after 24h at 21°C and 32% humidity. This is double the time of the THRIFTY protocol presented in the manuscript.

3: Homogenization after rehydration in water for 25 min at 21°C. Represents an extreme situation where freeze-dried muscle re-absorbs all the water previously removed during freeze-drying.

As can be seen in the example blots, there is no difference between the control sample and the sample left out for 24h. In contrast, proteolysis is clearly evident in the rehydrated samples.

We are therefore fully confident that there is no trade-off when using THRIFTY.

Assumably if 1000 fibres are collected to be analysed together, for the cost-time benefit outlined, what validation has been done to ensure that the fibre collection time does not affect downstream steps and measurements?

We are not fully clear on what the reviewer means here, but we never collect 1000 fibres at a time, nor do we state that in the manuscript. Fibre collection occurs prior to typing and the number of fibres collected during a day depends on several aspects, like the quality of the muscle biopsy and the skill of the technician. However, this applies to all methods and therefore has no relevance for assessing THRIFTY as a fibre typing method. Once fibres are ready for typing, THRIFTY is faster than dot blot independent of how many fibres were collected at the same time.

For clarification, what we state in the manuscript is that a skilled technician can type 800-1000 fibres under less than 11 hours. There is no rationale to expect any negative effects on downstream applications within this time, or even twice the time (see above), at least when using freeze-dried muscle fibres, as virtually no water is absorbed when fibres are handled according to standard procedure. This is evident from the proof-of-concept data included in the manuscript which all show expected differences and values within physiological range (i.e. higher cs activity and mitochondrial protein content in type I fibres, and lower glycogen levels in fibres from depleted muscle compared to fibres from a loaded muscle).

Also, if one has concerns about time-dependent negative effects for certain downstream measurements when using THRIFTY, it is easy to reduce the time by simply typing fewer fibres. There is no requirement to type 1000 fibres at the same time and THRIFTY still outperforms other contemporary methods when it comes to speed.

One of the issues in single fiber methodology is the lack of being able to simply and accurately detect MHC IIX fibers in humans. In this manuscript, this issue remains, however as pointed out by the authors, in muscle samples from healthy individuals this number would be low and so remain out of scope. For a similar reason, the number and type of hybrid fibers that are present, whilst with THRIFTY can be observed (assuming the microscopy method is used), yet once again the number would be low and so downstream measurements would be unlikely. As such, there is no benefit of THRIFTY to add these missing steps for single fiber analyses.

We fully acknowledge that THRIFTY has not been validated for type IIX fibres. However, we respectfully disagree with the reviewer's statement about THRIFTY not adding any value for measurements on hybrid fibres.

THRIFTY is currently the only method that can identify hybrid fibres with certainty as it does not rely on denatured samples. Under the denatured conditions used for SDS-PAGE, single fibre western blot and dot blot, it is impossible to distinguish true hybrid fibres from fibres contaminated by the opposing fibre type. This issue is eliminated with THRIFTY. As such, THRIFTY does in fact add great value with its improved accuracy for researchers interested in measurements on hybrid fibres.

Page 17: "Given these large variations, pooling too few fibres may prevent proper identification of mechanistic links between different variables." And "..... the pool should consist of as many fibres as possible."

Variation is an issue, however a number of fibers should be provided, not 'as many as possible'. One could go on collecting fibers for a very long time, so the number that 'is possible' is very high. Choosing '100' is random, some analyses should be done to arrive at the number of fibres required.

We agree with the reviewer's point that the number that is possible can be very high. However, this is a very general statement where we are trying to make the point that that in general terms, a large number of fibres will always be more representative than very few fibres, especially when there is uncertainty about the number of fibres required.

Providing such numbers in the manuscript is far beyond the current scope but we have rephrased this paragraph for more nuance. We do also provide several references for different variables in which variation has been noted, which can be used as starting points for researchers wanting to establish the minimum number of fibres required for their specific analyses.

We used 100 fibres of each type to prepare the control slide for the modified THRIFTY protocol to be used with the western blot equipment, but we do not state anywhere that 100 fibres should be used for subsequent analyses.

Page 23: "high resolution mitochondrial respirometry" would require fresh tissue, how could the fibres be collected, fiber typed and then used for measurements? This would still be a delay of some hours. In order to make this statement, data should be provided.

We have now performed respirometry measurements on pools of type I and type II fibres from freshly collected muscle. Please see figure 10.

Dissection, typing and pooling took approximately 6 hours and within this time frame there is no significant time effect on respiration when muscle is kept in cold preservation media (Cardinale et al. *Physiol Rep.* 2018 Feb;6(4):e13611)

Fig S4: appears to show data for a single set of type I or type II fibers. This is not representative, and would need a higher 'n' for each measure and statistics to be included.

The data presented are not meant to show any effects of an intervention but are only for proof-of-concept to show the strength of THRIFTY in generating large pools of fibres for a wide range of subsequent analyses.

Dear Dr Apró,

Re: JP-TFP-2022-283262XR1 "THRIFTY, a novel high-throughput method for rapid fibre type identification of isolated skeletal muscle fibres" by Oscar Horwath, Sebastian Edman, Alva Andersson, Filip J Larsen, and William Apró

I am pleased to tell you that your paper has been accepted for publication in The Journal of Physiology, subject to any modifications to the text and/or satisfactory clarification of the Methods section that may be required by the Journal Office to conform to House rules.

NEW POLICY: In order to improve the transparency of its peer review process The Journal of Physiology publishes online as supporting information the peer review history of all articles accepted for publication. Readers will have access to decision letters, including all Editors' comments and referee reports, for each version of the manuscript and any author responses to peer review comments. Referees can decide whether or not they wish to be named on the peer review history document.

The last Word version of the paper submitted will be used by the Production Editors to prepare your proof. When this is ready you will receive an email containing a link to Wiley's Online Proofing System. The proof should be checked and corrected as quickly as possible.

The accepted version of the manuscript is the version that will be published online ahead of the copy edited and typeset version being made available. Authors should note that it is too late at this point to offer corrections prior to proofing. Major corrections at proof stage, such as changes to figures, will be referred to the Reviewing Editor for approval before they can be incorporated. Only minor changes, such as to style and consistency, should be made a proof stage. Changes that need to be made after proof stage will usually require a formal correction notice.

All queries at proof stage should be sent to TJP@wiley.com

Are you on Twitter? Once your paper is online, why not share your achievement with your followers. Please tag The Journal (@jphysiol) in any tweets and we will share your accepted paper with our 22,000 plus followers!

Yours sincerely,

Scott K. Powers
Senior Editor
The Journal of Physiology
<https://jp.msubmit.net>
<http://jp.physoc.org>
The Physiological Society
Hodgkin Huxley House
30 Farringdon Lane
London, EC1R 3AW
UK
<http://www.physoc.org>
<http://journals.physoc.org>

P.S. - You can help your research get the attention it deserves! Check out Wiley's free Promotion Guide for best-practice recommendations for promoting your work at www.wileyauthors.com/eeo/guide. And learn more about Wiley Editing Services which offers professional video, design, and writing services to create shareable video abstracts, infographics, conference posters, lay summaries, and research news stories for your research at www.wileyauthors.com/eeo/promotion.

* IMPORTANT NOTICE ABOUT OPEN ACCESS *

Information about Open Access policies can be found here <https://physoc.onlinelibrary.wiley.com/hub/access-policies>

To assist authors whose funding agencies mandate public access to published research findings sooner than 12 months after publication The Journal of Physiology allows authors to pay an open access (OA) fee to have their papers made freely available immediately on publication.

You will receive an email from Wiley with details on how to register or log-in to Wiley Authors Services where you will be able to place an OnlineOpen order.

You can check if your funder or institution has a Wiley Open Access Account here <https://authorservices.wiley.com/author-resources/Journal-Authors/licensing-and-open-access/open-access/author-compliance-tool.html>

Your article will be made Open Access upon publication, or as soon as payment is received.

If you wish to put your paper on an OA website such as PMC or UKPMC or your institutional repository within 12 months of publication you must pay the open access fee, which covers the cost of publication.

OnlineOpen articles are deposited in PubMed Central (PMC) and PMC mirror sites. Authors of OnlineOpen articles are permitted to post the final, published PDF of their article on a website, institutional repository, or other free public server, immediately on publication.

Note to NIH-funded authors: The Journal of Physiology is published on PMC 12 months after publication, NIH-funded authors DO NOT NEED to pay to publish and DO NOT NEED to post their accepted papers on PMC.

EDITOR COMMENTS

Reviewing Editor:

The authors have satisfactorily addressed the concerns of the 3 reviewers. I am sorry that this has been a relatively long review, but congratulations for getting this done.

Senior Editor:

Thank you for your patience and congratulations on the completion of an excellent study.

REFeree COMMENTS:

Referee #1:

No further comments

Referee #5:

The authors satisfactorily addressed my concerns. Well-done. Thank you.

Referee #6:

The manuscript is improved with the points of clarity. The following further comments should be considered:

The 3x faster is post fiber collection, so would be better as an amount of time from start to finish for a certain number of fibers.

The data about water and humidity is worth including in the manuscript.

It is good that the authors have now shown that the muscle fibers could be fiber typed, pooled and used for respiratory measurements, however the authors need to discuss those findings, albeit n=1. Are they as expected? Why not pool a smaller number of fibers and show the reproducibility of the assay? This is necessary for those new data to be put into perspective with the literature, with the inclusion of the fiber typing step.